# Developing a drought monitoring index for the Contiguous U.S. using SMAP

Sara Sadri[Princeton University], Eric F. Wood[Princeton University], and Ming Pan[Princeton University]

[Princeton University]Department of Civil and Environmental Engineering, 59 Olden St, Princeton, NJ 08540

**Correspondence:** Sara Sadri (sadri@princeton.edu)

**Abstract.**

Since April 2015, NASA's Soil Moisture Active Passive (SMAP) mission has monitored near-surface soil moisture, mapping the globe (between $85.044°N/S$) using an L-band (1.4 GHz) microwave radiometer in 2-3 days depending on location. Of particular interest to SMAP-based agricultural applications is a monitoring product that assesses the SMAP near-surface soil moisture in terms of probability percentiles for dry and wet conditions. However, the short SMAP record length poses a statistical challenge for meaningful assessment of its indices. This study presents initial insights about using SMAP for monitoring drought and pluvial regions with a first application over the Contiguous United States (CONUS). SMAP soil moisture data from April 2015 to December 2017 at both near-surface (5cm) SPL3SMP, or Level 3, at ∼36 km resolution; and root zone SPL4SMAU, or Level 4, at ∼9 km resolution were fitted to beta distributions and were used to construct probability distributions for warm (May-October) and cold (November-April) seasons. To assess the data adequacy and have confidence in using short-term SMAP for drought index estimate, we analyzed individual grids by defining two filters and a combination of them, which could separate 5,815 grids covering CONUS into passed and failed grids. The two filters were: (1) The Kolmogorov-Smirnov (KS) test for beta-fitted long-term and short-term Variable Infiltration Capacity (VIC) LSM with 95% confidence; and (2) Good correlation ($\geq 0.4$) between beta-fitted VIC and beta-fitted SPL3SMP. To evaluate which filter is the best, we defined a Mean Distance (*MD*) metric, assuming VIC index at 36 km resolution is the ground truth. For both warm and cold seasons, the union of the filters – which also gives the best coverage of the grids throughout CONUS – was chosen to be the most reliable filter. We visually compared our SMAP-based drought index maps with metrics such as U.S. Drought Monitor (from D0-D4), SPI 1 month and VIC near surface from Princeton University. The root zone drought index maps were shown to be similar to those produced by the VIC at the root zone, SPI 3 month, and GRACE. This study is a step forward towards building a national and international soil moisture monitoring system, without which, quantitative measures of drought and pluvial conditions will remain difficult to judge.

## 1 Introduction

Drought is an extreme condition when water in one or a combination of water stores (e.g. river, lake, reservoir, snowpack, soil water, or groundwater) or water fluxes (precipitation, evapotranspiration or runoff) drops below a defined condition for a prolonged period of time (Wilhite and Glantz, 1985; Wilhite, 2000; AMS, 2012). Such a water deficit evolves over weeks to

months and can last for months and years. Drought's propagation is silent and often without warning until it impacts human lives and environmental activities (Tallaksen and Van Lanen, 2004). Drought conditions are related to water demand, so local water use plays a central role in defining conditions of scarcity and the resulting impacts. Wilhite and Glantz (1985) classified drought into meteorological, agricultural or hydrological depending on whether the deficit is measured using precipitation, soil moisture or river discharge, respectively.

The reduced supply of precipitation (and subsequently soil moisture) for crops leads to an agricultural drought that impacts crop yield, inflicting enormous economic impacts in developed countries and the suffering of millions of people in less developed regions of the world (Université Catholique de Louvain, 2009). In the U.S. since 1996, there has been at least one drought event per year except for years 1997, 2001, 2004, and 2010, and each year drought cost between 1 billion and 14 billion dollars in damages (in 2015 - adjusted dollars) (NOAA, 2018b). In California alone, the 2015 drought was estimated to cause $2-5 billion in damages to the agricultural sector (Howitt et al., 2015).

Although the impacts of drought are intimately linked to the vulnerability of a population to adverse conditions (UN/ISDR, 2007) and how society responds within the constraints of changing economies, timely determination of the current level of agricultural drought aids the decision-making process in order to reduce its impacts. Scientifically-based drought monitoring tools and warning systems assist to mitigate the losses caused by droughts and to plan and manage water shortages that will accompany future droughts (Martinez-Fernandez et al., 2016). Such drought monitoring tools are based on long-term observations of the hydrological variables such as precipitation, streamflow, soil moisture, and groundwater.

Pluvial conditions are related to an abundance of precipitation and subsequently wet soil conditions that can adversely affect agriculture by water-logging the fields or exacerbating flooding from additional rainfall. Thus, for monitoring extremes (either agricultural drought or pluvial conditions) realistic estimation of soil moisture at regional to continental scales is required. Soil moisture is the central source of information since it reflects recent precipitation and antecedent soil conditions (Sheffield and Wood, 2011). In a sense, soil moisture captures the aggregate balance of all hydrological processes and represents available water, being a buffer between incoming precipitation and throughfall, and evapotranspiration and drainage processes (Entekhabi et al., 1996). Unfortunately, soil moisture (and evapotranspiration) are among the least-observed components of the hydrological cycle, especially over large spatial and temporal scales (Reichle, 2017; Sheffield and Wood, 2011).

Many statistical measures, or indices, for extreme conditions, have been developed in the U.S.; particularly for drought conditions. This is due to the slow evolution of drought and its economic and social impact. Currently, no single drought index has been able to adequately capture the severity and intensity of drought and its impact on different groups of users (Heim, 2002). Heim (2002) gives an overview of the major twentieth-century U.S. drought indices. The most common ones are the Standardized Precipitation Index (SPI), Palmer Drought Severity Index (PDSI), Standardized Runoff Index (SRI) and the U.S. Drought Monitor (DM or USDM).

SPI is recognized by the World Meteorological Organization (WMO) as the standard index for quantifying and reporting meteorological drought. It is used to characterize drought on a range of time scales from 1 to 36 months. The raw precipitation is fit to an appropriate distribution function, then transformed into a standardized Normal distribution. The SPI index is expressed as the number of standard deviations by which the anomaly deviates from the long-term mean. On short timescales, SPI is

closely related to soil moisture while at long timescales it is related to groundwater. The advantages of SPI include that: it only relies on precipitation; it can characterize both drought and pluvial conditions; its computation over different timescales can be related to various water resource stores (such as soil moisture and groundwater); and it is more comparable across regions with different climates than the Palmer Severity Drought Index (PDSI). The key limitations of SPI are: it is sensitive to the quantity of the data used. Usually, 30 years of monthly precipitation data is recommended for fitting the data. Additionally, SPI is a meteorological tool that measures water supply but does not account for evapotranspiration. This limits its ability to capture the effect of increased temperatures (associated with climate change) on moisture demand and availability. Finally, SPI does not consider the intensity of precipitation and how it impacts on runoff and streamflow. Overall, SPI can inform about anomalies in precipitation so it needs to be used in combination with other information in order to be useful for agricultural drought assessment (NCAR, 2018).

The PDSI uses precipitation and an estimate of evaporation in conjunction with a water balance model to estimate relative soil dryness/potential evapotranspiration. The original formulation used only the temperature to estimate a potential evapotranspiration, but it is now recognized that an energy-based approach, such as the Penman-Monteith approach, is preferred (Sheffield et al., 2012; Mo and Chelliah, 2006). Since PDSI uses potential evapotranspiration and precedent (prior month) conditions, it takes into account the basic effect of global warming and is effective in determining long-term drought, especially over low and middle latitudes. Key limitations of PDSI include that PDSI is not as comparable across regions as the SPI and lacks the ability to handle winter-time conditions that include snowmelt and frozen precipitation, which makes its long-term monitoring problematic. Unlike SPI indices, PDSI lacks multi-timescale features, making it difficult to correlate with specific water resources like runoff, snowpack, and reservoir storage.

SRI is based on SPI and a model runoff. The strength of SRI, as a runoff-based index, is that it can be used to forecast future runoff and its predictability depends not only on climate outlooks, for which seasonal skill is generally low, but also on hydrologic initial conditions (e.g., spring snow state in the western US). The disadvantage of SRI is similar to the disadvantage of using any modeled runoff: since modeled runoff cannot be verified everywhere, the runoff-based indices of SRI reflect the customary uncertainties associated with model outputs (Shukla and Wood, 2008).

The USDM integrates several drought indices and professional input from all levels into a weekly operational drought-monitoring map product (Svoboda, 2000). The limitation of the USDM lies in its attempt to show drought at several temporal scales (from short-term drought to long-term drought) on one map product. Hence, the application of the DM is not to replace any local or state information or subsequently declared drought emergencies or warnings, but rather to provide a general assessment of the current state of drought around the United States, Pacific possessions, and Puerto Rico (Svoboda, 2000). Since the USDM relies on professional inputs from the field, it is difficult to have historical consistency (since the professionals change) or to provide forecasts.

Long-term and large scale observations of soil moisture are scarce in the United States and elsewhere, so datasets produced by the North American Land Data Assimilation System (NLDAS) are valuable alternatives. Currently National Centers for Environmental Prediction (NCEP) offer an NLDAS drought monitor (NOAA, 2018a) based on four land surface models (LSMs): VIC, Noah, Mosaic, and Sacramento. Sheffield et al. (2004) used simulations from the NLDAS Variable Infiltration Capacity

(VIC) model forced with observed precipitation and near surface meteorology to develop a soil moisture based drought index. The approach (Sheffield et al., 2004) took was to fit the VIC-simulated soil moisture to probability distributions, usually beta distributions, where the percentiles are translated to the index values that range from 0 to 1. Recent drought applications such as the VIC-based Princeton University drought and flood monitoring systems for Africa and Latin America (Sheffield et al., 2014) use the simulated soil moisture, which is mostly based on satellite precipitation (Princeton University Hydrology, 2013).

A major limitation of the indices discussed earlier, as well as the LSM based approaches, is a reliance on quality meteorological data. While precipitation is one of the best observed variables, gauge observations are limited in many regions, especially over much of the developing world. Even when they are available, they are often not in near real time, preventing computing indices. This reveals one of the weaknesses of the above indices: their estimates rest on the availability and accuracy of the forcings, specifically precipitation (Reichle, 2017). In places such as the U.S. where the quality of the precipitation data is quite high, VIC quality is relatively high also (Pan et al., 2016). However, in regions with sparse networks or low accessibility, such as Africa, the VIC quality can be relatively low (Reichle, 2017). Additionally, intercomparison of the four NLDAS models showed that soil moisture differs considerably among models (Robock et al., 2000).

Heim (2002) summarizes four characteristics of a useful operational drought monitoring system. These include: 1) the indices need to be available on a near-real-time basis; 2) the indices need to be monitored on a national scale, which will require the establishment of national networks for some variables; 3) a complete and reliable historical data are needed over a common reference period to allow conversion of the observations to a meaningful form (such as a percentile ranking); and 4) the data need to be adjusted to remove nonclimatic influences (such as those arising from water management practices) (Friedman, 1957; Heim, 2002).

An alternative approach to using model-derived soil moisture for drought detection and prediction is satellite-derived soil moisture. There are currently four major satellite-based systems that provide soil moisture products at various spatial and temporal resolutions: MetOp with the advanced scatterometer (ASCAT) (Brocca et al., 2010; Wagner et al., 2013), JAXA's Advanced Microwave Scanning Radiometer AMSR2 (Parinussa et al., 2015; Wu et al., 2015) with the C- and X Band passive radiometers on the GCOM-W1 satellite that is a follow-on to the AMSR-E sensor, which failed on 4 October 2011 and was part of NASA's Earth Observing System; ESA's Soil Moisture Ocean Salinity (SMOS) L-band radiometer (Pan et al., 2010; Kerr et al., 2012, 2016) and NASA's Soil Moisture Active Passive (SMAP) L-band radiometer Entekhabi et al. (2010). The radar on SMAP failed after three months, but soil moisture estimates based on the radiometer continue to be produced.

Of particular interest, especially for applications in parts of the globe with sparse in-situ data, is to have a SMAP-based monitoring product that expresses soil moisture in terms of probability percentiles for dry (drought) or wet (pluvial) conditions (Entekhabi et al., 2010). This study presents insights and the potential of using SMAP for monitoring drought and pluvial regions with a first application over the Contiguous United States (CONUS). We fit the soil moisture data from SMAP at both the level 3, 5 cm passive radiometer retrievals (SPL3SMP) and the level 4, root zone product that assimilates the surface SPL3SMP into the Catchment land surface model (SPL4SMAU) to beta distributions and construct probability distributions for warm and cold seasons, and measure the reliability of our estimates. Producing soil moisture drought indices at two different soil depths allow for monitoring of agricultural drought in different stages of development (NDMC, 2018a). This is important,

first, because grid analysis showed that full column soil moisture index can be less, similar, or more than near surface soil moisture index. Secondly, depending on the plant development stage, surface soil moisture or root zone soil moisture drought index can be more useful in agricultural management. For example, surface soil moisture is important in the germination stage but less so for managing irrigation or in estimating yields. Deficient topsoil moisture at planting may hinder germination, leading to low plant populations per hectare and a reduction of final yield (NDMC, 2018a). At the same time root zone moisture at this early stage may not affect final yield but as the growing season progresses it becomes more important for plant water needs.

The rest of this paper as follows: the SMAP data is discussed in section 2.1, including a determination whether its 1,006 days are sufficient to estimate a drought index. Section 2.2 develops the indices by fitting beta distributions, with upper and lower bounds, to the time series and using the percentiles as the index. Section 2.3 develops a numerical analysis of the adequacy of the SMAP data. In section 3, results of adequacy tests are discussed and comparisons are made to the currently available drought indices. To help relate the percentiles to the U.S. Drought Monitor, which uses levels D0-D4 to indicate severity, the percentiles are mapped. We also extended our indices to pluvial conditions similar to the maps from GRACE and Princeton University. Conclusions are brought in sections 4.

## 2 Data and Methods

### 2.1 SMAP Data

Since April 2015, NASA's SMAP mission has been monitoring near-surface soil moisture, mapping the globe (between $85.044°N/S$) using an L-band (1.4 GHz) microwave radiometer in 2-3 days depending on location. The SMAP mission provides a set of operational global data products that include:

- Level 3 (SPL3SMP): a composite based on daily passive radiometer estimates of global land surface soil moisture (nominally 5 cm) that are resampled to a global, cylindrical 36 km Equal-Area Scalable Earth Grid, Version 2.0 (EASE-Grid 2.0) (O'Neill et al., 2016). For this study, version 4 of SPL3SMP is used, which is the release version from the very beginning of the launch of SMAP. The release number changes over time. R16 version is the latest version released in June 2018. However, in all release versions of SMAP, including version 4, regions with permanent snow/ice, frozen ground, excessive static or transient open water in the cell or excessive radio-frequency interference (RFI) in the sensor data, and heavy vegetation (vegetation water content $> 4.5 \ kg/m^2$) are masked out using a Normalized Polarization Ratio (NPR)-based passive freeze-thaw retrieval. Given the 1000-km swath and 98.5-minute orbit, the SPL3SMP retrievals are spatially and temporally discontinuous with 2-3 day gaps depending on location; and

- Level 4 (SPL4SMAU): provides global surface and root zone soil moisture by assimilating the SMAP L-band brightness temperature data (for which SPL3SMP is the gridded version) from descending and ascending half-orbit satellite passes, approximately 6:00 a.m. to 6:00 p.m., every 3 hours, local solar time, into NASA's Catchment LSM (Reichle, 2017; Reichle et al., 2015). The SPL4SMAU data product is gridded using an Earth-fixed, global, cylindrical 9 km EASE-Grid

2.0 projection. The land surface model component of the assimilation system is driven by a forcing data stream from the global atmospheric analysis system at the NASA GMAO (Rienecker and coauthors, 2008). Additional corrections are applied using gauge- and satellite-based estimates of precipitation that are downscaled to the temporal and 9 km scale of the model forcing using the disaggregation methods described in Liu et al. (2011) and Reichle et al. (2011). The SPL4SMAU product provides global soil estimates for the surface (0-5 cm) and "root zone" (0-100 cm) and is an effort to provide continuous, daily information without discontinuous data restrictions due to gaps in the SPL3SMP soil moisture retrievals. Nonetheless, the only product that doesn't use ancillary meteorological data is the SPL3SMP soil moisture retrievals.

In this study, SPL3SMP products from the 6:00 a.m. retrievals and SPL4SMAU products from 6:00 a.m. retrievals, are used in the analysis of soil moisture drought index. Our SMAP data records are from 2015-04-01 to 2017-12-31, which is equivalent to 1,006 days.

The approach selected here is somewhat similar to that from Sheffield et al. (2004) where the soil moisture time series are fit to a beta distribution (with upper and lower bounds) and the distribution percentiles are the index values. There are, however, differences in our approach from that in Sheffield et al. (2004). Firstly, the basis of the data used in Sheffield et al. (2004) was simulated soil moisture from VIC while ours is remotely sensed data. Secondly, to calculate the bounds of beta distribution [a, b], Sheffield et al. (2004) used the first (last) 10% of the sorted soil moisture values linearly related to the empirical cumulative distribution function. In our study, this approach did not yield useful results with the estimated limits for a (b) for SMAP, often did not cover the full range of observed values, preventing interpretation of the historical data. Our methodology for obtaining beta distribution parameters a and b are discussed in this section.

As mentioned in the introduction by Heim (2002), one of the conditions for an index approach is a complete and reliable historical data needed over a common reference period to allow conversion of the observations to a meaningful form. The short SMAP record length of 1,006 days, from 2015-04-01 to 2017-12-31, provides a statistical challenge in estimating the drought and pluvial indices, and thus the reliability assessments related to these extreme conditions are necessary. Therefore, to assess the data adequacy, we used a 1979-2017 VIC LSM simulation over CONUS. The VIC runs were carried out at a 4 km spatial resolution, and for the SPL3SMP comparisons averaged up to 36 km. Here we refer to it as VIC near surface (VIC-ns). The SPL4SMAU is at 9 km spatial resolution, so VIC data were aggregated from 4 km computing grids, and averaged over 3 soil layers with varying total soil thickness. We refer to it as VIC root zone (or VIC-rz). A statistical comparison is made between fitting a beta distribution to the VIC soil moisture values using only days when SPL3SMP soil moisture retrievals are available and for the complete 1979-2017 VIC data record. The Kolmogorov–Smirnov (KS) statistical test was used to evaluate the consistency of the beta fitted data. We made the assumption that grids that passed the consistency test using VIC data – i.e. the distribution from the SMAP period record and the complete record were deemed statistically the same – then the SMAP time series over that grid was sufficient to provide an index. More discussion of these results is given in section 3.

Furthermore, we looked at the frequency distribution of soil moisture data at each grid. The data seemed to be dominated by low soil moisture in the summertime, and high soil moisture in the wintertime. Therefore, to capture this inter-seasonal behavior in soil moisture, we divided the record into a warm season (April - September) and a cold season (October - March). Dividing

the year into warm and cold seasons enabled us to track the soil moisture dynamics, and thus the probability distribution and index seasonally. Ideally, we would have divided it into monthly data but there are insufficient observations.

For our study period, each grid has between 144 and 329 SPL3SMP soil moisture retrievals during the warm season and from 16 to 272 retrievals during the cold season. Figure 1 shows that the number of overpasses per grid is related to the latitude, with
higher latitudes having a higher number of overpasses, and to the season, with fewer values retrieved during winter, especially in the western U.S., due to snow cover and frozen ground. For LSPL4SMAU root zone, there are 457 records for the cold season and 549 records for the warm season for each grid.

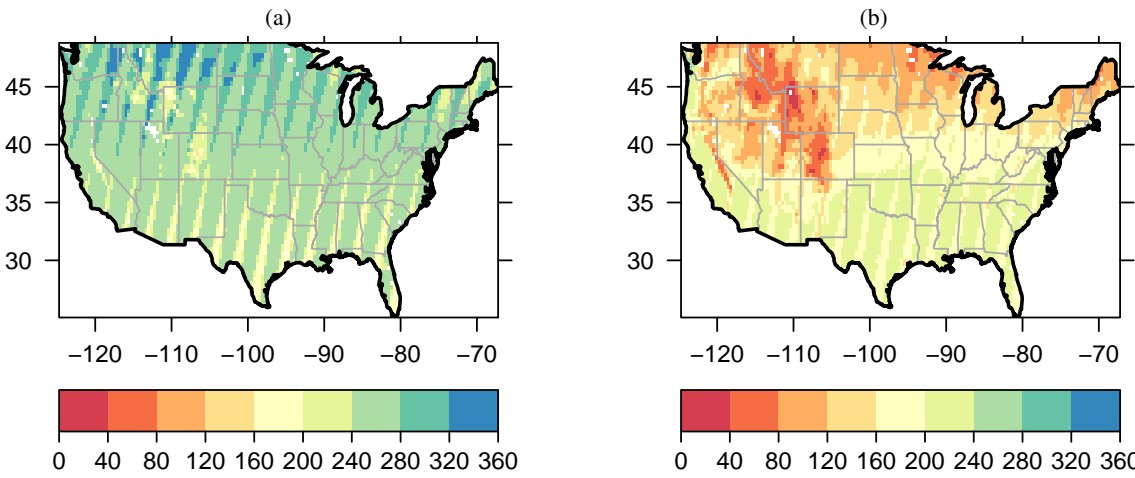

**Figure 1.** Number of retrievals for each season. (a) is warm season April 1 - September 30; (b) is the cold season, October 1 - March 31).

## 2.2   Fitting the beta distribution to the SMAP time series

The beta distribution is a family of continuous distributions with two shape parameters ($p$ and $q$). It generalizes to a bounded
distribution on the interval of $[a, b]$, where $a$ and $b$ usually take on the values of 0 and 1. The beta distribution is flexible enough to model a wide variety of shapes. In our study, we compared the beta distribution to several parametric distributions (including Normal and Gumbel), but the beta distribution showed the best goodness-of-fit. Furthermore, given the bounded nature of the distribution, it is often used as the model of choice for modeling soil moisture time series (Sheffield et al., 2004). The general formula for the beta probability density function (pdf) is:

$$f(x) = \frac{(x-a)^{(p-1)}(b-x)^{(q-1)}}{B(p,q)(b-a)^{p+q-1}} \quad a \leq x \leq b; \quad p, q > 0 \tag{1}$$

where $p$ and $q$ are shape parameters, $a$ and $b$ are lower and upper bounds, respectively of the distribution. In case where $a = 0$ and $b = 1$, this becomes a standard beta distribution (NIST, 2013). $B(p,q)$ is a beta constant computed with the formula

$$B(p,q) = \int_0^1 t^{p-1}(1-t)^{q-1}dt \tag{2}$$

A main challenge is to fit the four parameters of the beta distribution, given a set of empirical observations. Sheffield et al. (2004) used the method of moments to fit the beta distribution to historical soil moisture simulations from the VIC LSM. They computed the first three moments and minimized the difference between the distribution estimates and sample estimates since they were over-constrained. We also used the standard method of moments to calculate the parameters $p$ and $q$. But for each grid location, we fit the beta distribution to 6 sets of data related to the SPL3SMP product: 1) Short warm season VIC and 2) Short warm season SMAP (1 April - 30 September for 2015, 2016, 2017; 18 months); 3) Long warm season VIC (1 April - 30 September, 1979-2017; 129 months); 4) Short cold season VIC and 5) Short cold season SMAP (1 October - 31 March, 2015-2016; and 1 October - 31 December 2017; 15 months); 6) Long cold season VIC (1 October - 31 March for 1979 and 2016; and 1 October - 31 December for 2017; 126 months), using the first and second moments $\mu = \frac{p}{p+q}$ and $CV = \frac{\mu}{\sigma}$, where $p$ and $q$ are parameters and its standard deviation defined as:

$$\sigma = \sqrt{\frac{p*q}{(p+q)^2*(p+q+1)}} \tag{3}$$

For the SPL4SMAU root zone soil moisture product, the beta distribution was fit to the warm season and cold season using all 457 and 549 records, respectively.

Figure 2 shows the 20th percentile, average and 80th percentile soil moisture data in the warm season and cold season for SPL3SMP 5-cm soil moisture product, and similarly in Figure 3 for the SPL4SMAU root zone product after data were fit to the beta distribution.

## 2.3 Data Adequacy Filters

Insufficient SMAP record length may result in unreliable index values. To be meaningful in using short SPL3SMP data for making confident predictions, we will analyze which grids have the highest certainty in our SMAP drought index. That is, we perform adequacy analysis, and defining filters that separate grids with high reliability in drought monitoring and prediction from ones where we don't expect our predictions to be as accurate. We first define two filters which can separate the 5,815 grids covering CONUS into grids that passed and failed quality control. The two filters are:

1. The Kolmogorov-Smirnov (KS) test for beta-fitted long-term and short-term VIC with 95% confidence;

2. Good correlation ($\geq 0.4$) between beta-fitted VIC and beta-fitted SPL3SMP.

Below we expand upon these two filters and then show how we used them to numerically find the best SPL3SMP filter. We also investigate if combinations of the filters are superior to the individual filters taken alone.

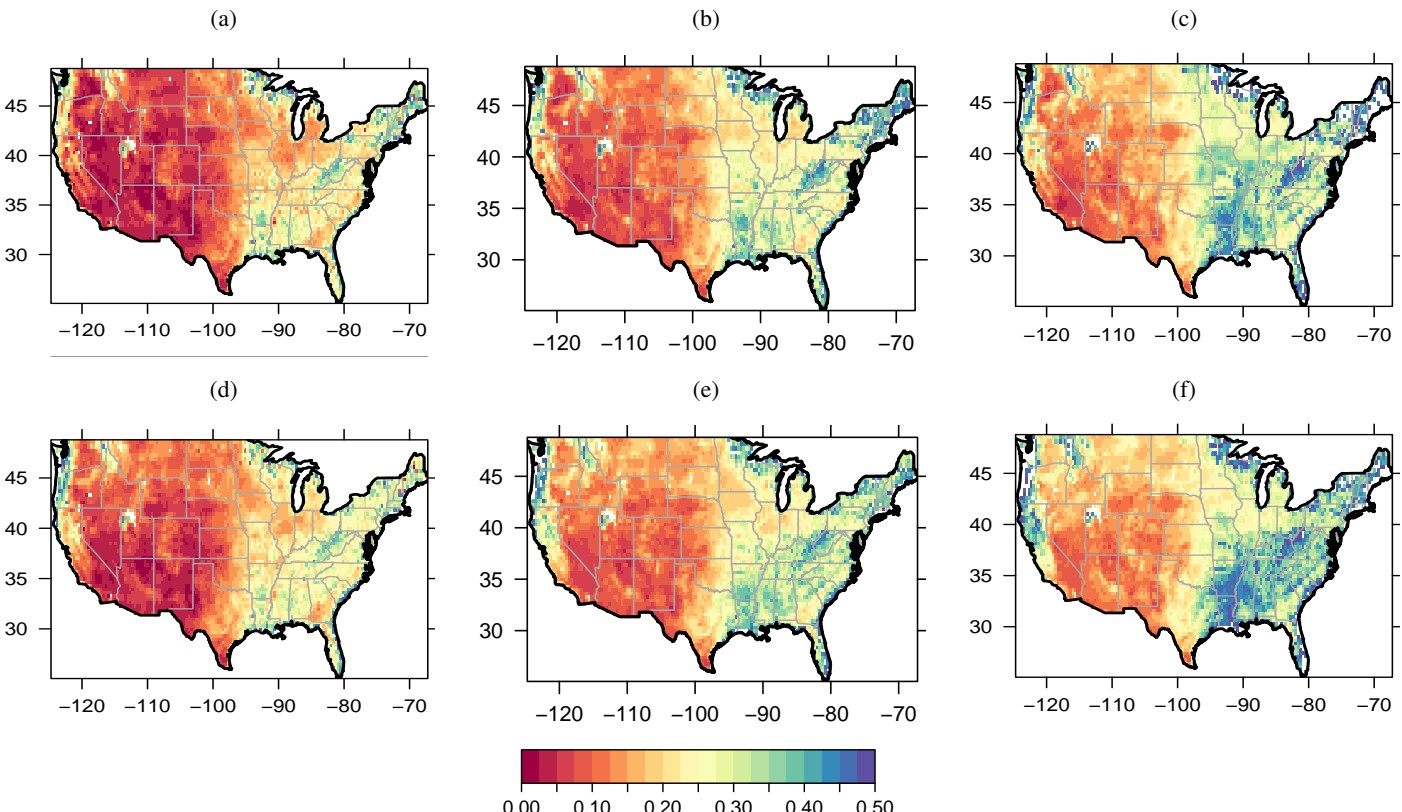

**Figure 2.** top row: SMAP soil moisture values for the warm season during summer for SPL3SMP top 5 cm soil moisture (a), 20th percentile; (b), average soil moisture; (c), 80th percentile; bottom row: same as the top row but for the cold season. Total period is from 2015/04/01 to 2017/12/31. The soil moisture unit is $m^3/m^3$.

### 2.3.1 Kolmogorov-Smirnov (KS) filter

The KS test is a well-known nonparametric statistical test that compares whether two samples are coming from the same continuous distribution. We used the KS test for each grid, comparing the modeled beta distribution of the long-term VIC with the modeled beta distribution of the short-term VIC, in both warm and cold seasons. This shows if the long-term and short-term distributions are statistically indistinguishable. If this strong condition is satisfied for a grid, then it is reasonable to assume for that grid that the short SMAP time series would be consistent with a hypothetical long SMAP time series. The null hypothesis – that the underlying beta distribution of short-term soil moisture data is the same as the underlying beta distribution of long-term soil moisture data for VIC – is rejected for values of the KS statistic $D$ that exceed a critical value at the 95% significance level: $D_{critical} = \frac{1.36}{\sqrt{n}}$ where $n$ is the number of observed variable (Lindgren, 1962).

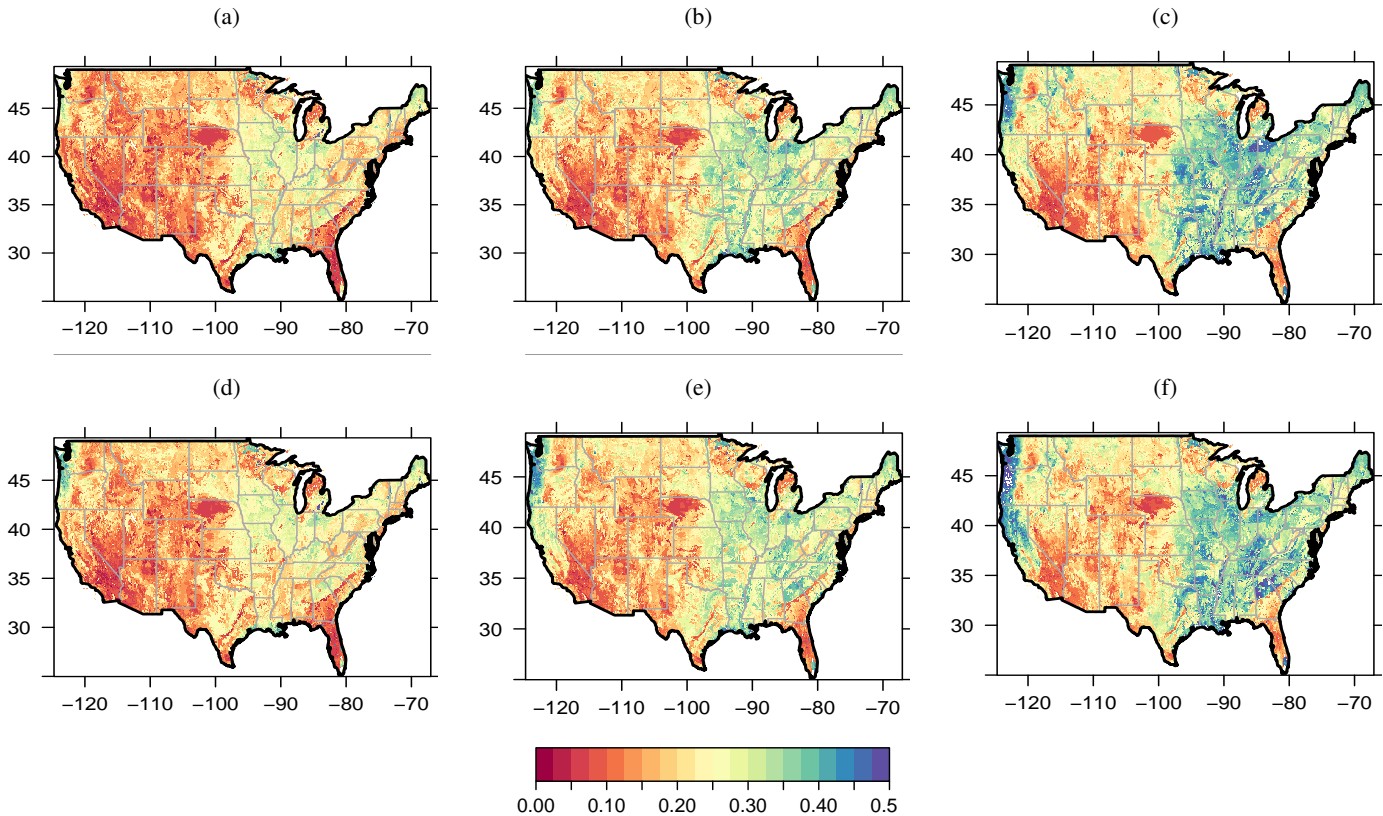

**Figure 3.** Same as shown in figure 2 but for SPL4SMAU (root zone soil moisture).

### 2.3.2 Correlation Filter

As mentioned earlier, one of the key assumptions of this paper is that if the beta distribution fit to the short-term VIC series is statistically consistent with beta fit to the long-term VIC time series, then we assume that the short-term beta-fitted SMAP series is consistent with the hypothetical long-term beta-fitted SMAP time series. This is possible because VIC modeled soil moisture is validated by ground measurements (Pan et al., 2016; Cai et al., 2017), and it is most plausible where the correlation between SPL3SMP and VIC is highest. Correlation maps are shown in Figure 4 between SPL3SMP and VIC-ns product for the warm season and cold season periods. This suggests another filter to use: require that the correlation of beta-fitted SPL3SMP and beta-fitted VIC soil moisture be relatively high. We examined the distribution of correlation values across all grids in order to pick the cutoff between high and low correlation. We chose the mean correlation, minus the standard deviation of correlation (across all grids), as a threshold. Thus grids whose correlation is close to average or better than the average pass the filter. For both the warm and cold seasons, this value was very close to 0.4 and as a result, we picked this as the common threshold.

### 2.3.3  Mean Distance (*MD*)

To evaluate whether the KS-based filter, the correlation filter, or a combination of both is best, we define a simple Mean Distance (*MD*) metric. Assuming VIC index at 36 km resolution is the ground truth, we can calculate a distance between VIC and SMAP. For every day that SMAP provided a retrieval, if $smap_i$ is the drought index percentile of grid $i$ that passes the
filter, and $VIC_i$ is the VIC drought index percentile of the same grid, and in total $n_g$ grids on day $d$ passed the filter, then the mean distance $MD_d$ is defined as the average of absolute distances between the SPL3SMP drought index percentiles and the VIC drought index percentiles. For the candidate date d and for a given filter:

$$MD_d = \frac{\sum_{i=1}^{n_g} |VIC_i - smap_i|}{n_g} \tag{4}$$

In equation (4), $VIC_i$ and $smap_i$ are VIC and SMAP drought index values for grid $i$, $n_g$ is the total number of grids that passed
the filter, and $MD_d$ is the mean distance for date $d$.

For each filter the final pass and fail distance scores are calculated by averaging $MD_d$ values over the number of days, especially for both dry or wet seasons:

$$MD = \frac{\sum_{i=1}^{n_d} |MD_d|}{n_d} \tag{5}$$

where $n_d$ is the total number of days for which the $MD_d$ value is available. While $n_g$ varies every day, since the number
of overpasses varies every day, the value of $n_d$ was constant (549 for warm season and 457 for cold season). The *MD* value obtained from grids failed a filter is called $MD_{fail}$ and the $MD$ value from grids passed a filter is called $MD_{pass}$. For each filter a difference (*Diff*) was computed by reducing the $MD_{pass}$ from the $MD_{fail}$: $Diff = MD_{fail} - MD_{pass} > 0$

### 2.3.4  Combination filters

In addition to the KS filter and the correlation filter, we investigate two filters defined by the following combination rules:

– Intersection filter: a grid cell $g$ passes the intersection filter if it passes both the KS filter *and* the correlation filter. Otherwise, it fails;

– Union filter: A grid cell $g$ passes the union filter if it passes *either* filter, or both. Note that using the union filter gives the best coverage of the grids throughout CONUS, while the intersection filter has the strongest requirements for passing.

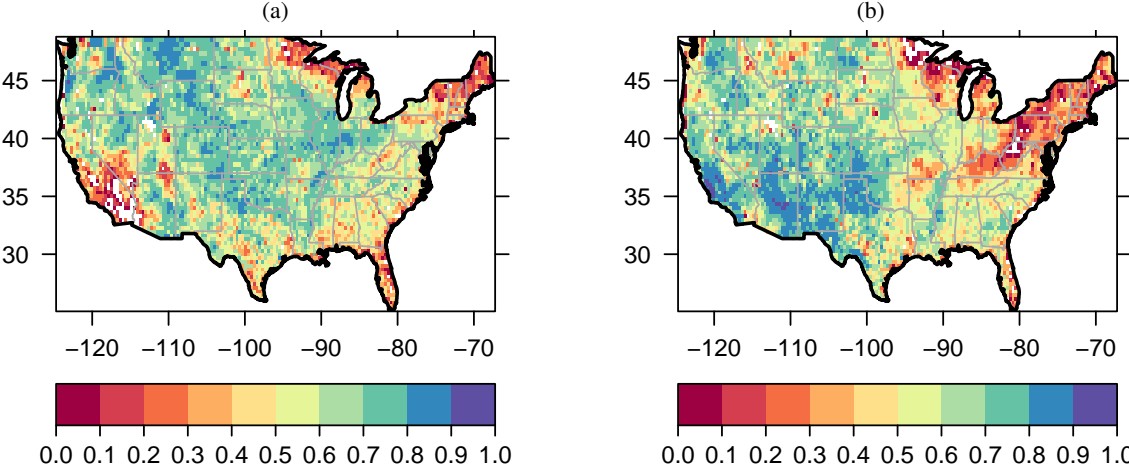

**Figure 4.** (a) Correlations (R) between VIC and SMAP beta models for the warm season (average R=0.57) and (b) cold season (average R=0.56). White regions signify a negative correlation.

## 3    Results and Discussion

### 3.1    Data adequacy metrics

#### 3.1.1    Correlation filter

Figure 4 shows that the average correlation for both warm and cold seasons are high and around 0.6. During the warm season,
the Central Valley and Southern California, Florida, northeastern U.S., and north of Wisconsin, and Minnesota show poor correlation with VIC, around 0.2. The extent of this poor correlation increases during the cold season for northeastern U.S., Wisconsin, and Minnesota. Snow season results in poor SMAP coverage during winter time in those areas. In addition, the low number of overpasses (presented in Figure 1) during winter in the northeast can play a role in the low amount of data and poor correlation during the cold season. Contrary to the warm season, southern California shows a high a correlation with VIC
during the cold season, around 0.9. We attribute this change in southern and south-central California from cold season to warm season to irrigation that SMAP picks up (Lawston et al., 2017), but VIC doesn't since the version used here doesn't have water management effects. Land use/land cover map shows that about one-third of these areas are irrigated vegetation and another third is forests and woodlands (USGS, 2018). There are also as many as 2 million water wells in California that contribute to the irregularity of groundwater and affecting the soil moisture. They range from hand-dug, shallow wells to carefully designed
large-production wells drilled to great depths (California Dept. of Water Resources, 2018). More data is needed before we can recognize further attributions to the low correlation between VIC and SMAP in that region. While systematic biases do not get revealed in correlations, the temporal consistency among the time series is captured.

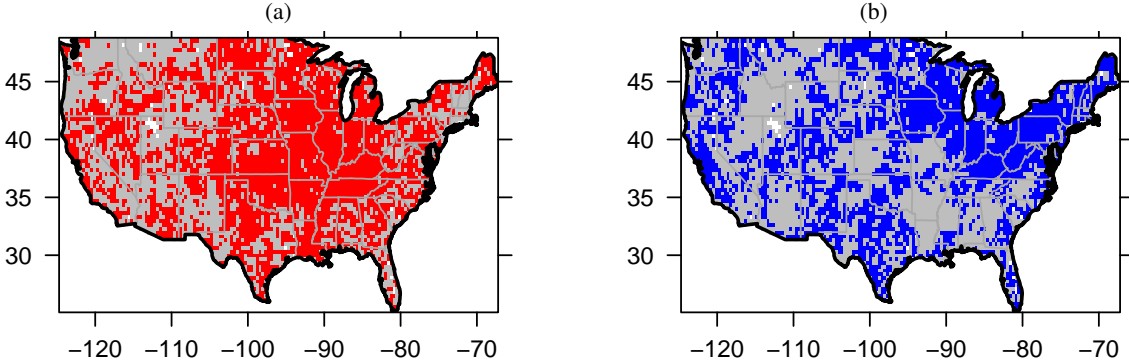

**Figure 5.** (a) Grids in red show areas whose short-term VIC in warm season data has the same underlying beta distribution as the long-term VIC in warm season data (n = 3560 or 68% of grids are red); (b) the same as the left figure but for cold season period shown in blue (n = 2927 or 57% of grids). Gray areas are grids where the short-term VIC does not have the same beta distribution as their long-term VIC.

### 3.1.2 KS filter

Figure 5 shows which grids passed the 95% KS test: there, we have confidence that the SMAP drought (pluvial) indices provide reliable risk levels given the current period of record. The warm season shows 11% more grids passing the adequacy test than the cold season. Note that as the record length gets extended, the above analysis needs to be repeated to see if the adequacy

changes.

In the warm season, the majority of the grids whose underlying short-term and long-term beta distribution were different were in the western U.S. The low warm season correspondence in the Pacific Northwest region is particularly apparent. The PNW region is covered by dense forests, mountains, and heavily regulated agricultural lands by irrigation. This contributes to the fact that most grids in PNW do not pass the KS filter. A pattern of low correspondence over the major mountain areas

(e.g. the Rockies, Sierra-Cascades) is also apparent, given the coarse SMAP brightness temperature (Tb) footprint and dense vegetation.

### 3.1.3 Combined filters

Figure 6 represents the results of Correlation filter and KS filter together for both warm (top figure) and cold (bottom figure) seasons over all 5,815 grids. We use these filters (passed/failed grids) on a daily basis for $MD_d$ measures; though the value

changes every day depending on the number of overpasses for that date. Table 1 summarizes how many grids pass or fail each filter.

### 3.2 Evaluation of Results Under Different Filters

For each filter, the values of $MD_d$ were averaged to calculate $MD_{fail}$ and $MD_{pass}$ for all CONUS over the 549 days of warm season and 457 days of cold season. The summary result of all 4 tests is shown in Table 2 and Table 3. To test if having a filter

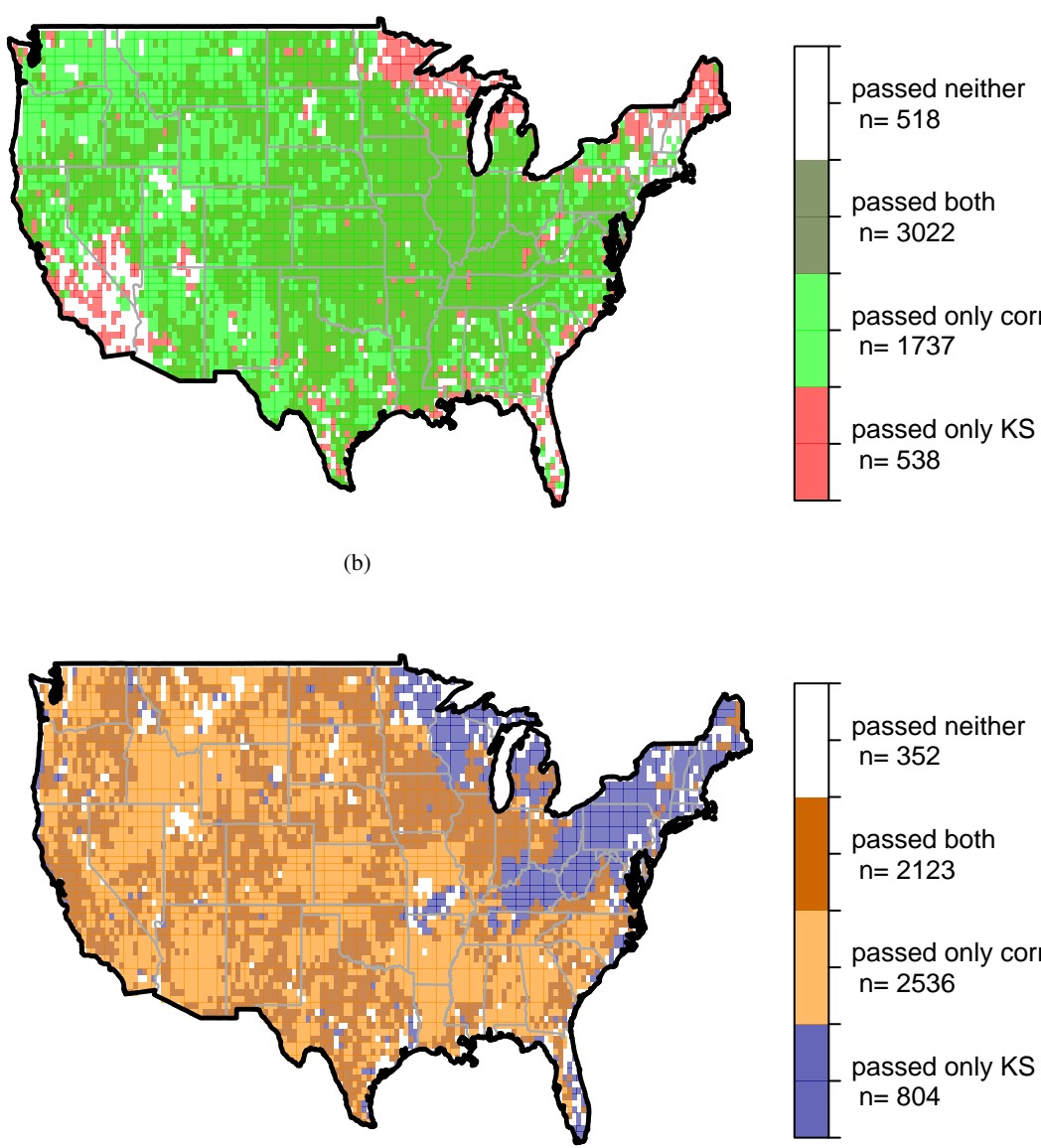

**Figure 6.** (a): warm season grids that pass the correlation filter and/or the KS filter. Dark green grids include grids that pass intersection filters. (b): cold season grids that pass the correlation filter and/or the KS filter. Dark orange grids include grids that pass intersection filters. In both figures, white grids show the grids that pass neither filter and will be crossed hatched in index maps.

is better than having no filter, for each season, we performed two-sided null hypothesis. The tests used 95% confidence limits between the *MD* of all grids – which was 22.7 in warm season and 22.6 in cold season – versus the $MD$ of only passed grids.

**Table 1.** Number of grids, out of total 5,815, that fail and pass the quality control for each filter.

| $n_g$ | KS filter | Correlation filter | Intersection filter | Union filter |
|---|---|---|---|---|
| Warm season fail | 2,255 | 1,056 | 2,793 | 518 |
| Warm season pass | 3,560 | 4,759 | 3,022 | 5,297 |
| Cold season fail | 2,888 | 1,156 | 3,692 | 352 |
| Cold season pass | 2,927 | 4,656 | 2,123 | 5,463 |

Note: Per day, the $n_g$ numbers are less because of SMAP overpass missing grids.

**Table 2.** $DS$ of four tests averaged over 549 days of warm season.

| | KS filter | Correlation filter | Intersection filter | Union filter |
|---|---|---|---|---|
| $MD_{fail}$ | 24.1 | 26.5 | 24.5 | 26.8 |
| $MD_{pass}$ | 21.9 | 21.9 | 21.1 | 22.3 |
| $Diff$ | 2.2 | 4.5 | 3.4 | 4.5 |

The results showed that all four filters are significantly different than the *MD* of all CONUS. Thus, regardless of the type of the filter, having some sort of filter is better than having no filter.

In warm season, the KS filter did better (i.e. larger *Diff* values, or better skill in separating high/low performance grids) than the correlation filter for only 115 days out of 546 days, mostly in April. For almost half of the dates (260 days out of 546), the union filter did better than the correlation filter. This outperformance of the union filter occurs evenly throughout the warm season.

In the cold season, for only 48 days out of 457 days, the KS filter did better than the correlation filter and for 198 days the union filter did better than the correlation filter. These results suggest that for the cold season, the correlation filter is providing the most effective filter. However, if we only accept the grids that pass the correlation filter, we lose 804 grids. This area involved almost all of the northeast coast and midcoast, as well as northern Wisconsin and northeast Minnesota. Although this is not a concerning problem for drought since most of the cold season these areas are covered by snow. We still decided to generate a cold season filter by including the KS filter with the correlation filter, thus we used the union filter for the cold season.

Three considerations for doing so are:

1. The Diff values: The correlation filter *Diff* value and union filter *Diff* Value during cold season are similar and close;

**Table 3.** $DS$ of four tests averaged over 457 days of cold season.

|  | KS filter | Correlation filter | Intersection filter | Union filter |
|---|---|---|---|---|
| $MD_{fail}$ | 22.8 | 29.0 | 24.1 | 29.2 |
| $MD_{pass}$ | 22.4 | 21.2 | 20.1 | 22.1 |
| $Diff$ | 0.4 | 7.8 | 4.0 | 7.1 |

2. The nature of our tests: It is not that surprising that the correlation filter has a higher $Diff$ than that from union filter. The $MD$ metric measures how the SMAP index resembles the VIC index. Thus, we find that the most important predictor is that the SMAP values should be correlated with the VIC values.

3. Optimum coverage: Although the cold season east coast drought index is not a matter of concern for this study, cold season soil moisture variability can affect warm season soil moisture and consequently agricultural drought. The goal is to create a filter that does not lose important information while provides the best knowledge of soil moisture data.

During the warm season, most of the grids that failed the test were in southern California and southern Nevada, in the northeast (New Hampshire, Massachusetts, and Connecticut), and in the southeast along the east coast of Florida. These are attributed to both the lack of correlation between SMAP and VIC and high variability between short-term and long-term soil moisture. These areas show non-stationarity in soil moisture meaning that soil moisture distribution is subject to change over time either due to climate or human interventions. During the cold season, most of the areas are covered using the union filter. However, as discussed we use this filter with caution knowing that at least according to our numerical analysis, the correlation filter did better than the union filter. The Great Lakes region, Minnesota, and Mid Atlantic Region do not show a high correlation between VIC and SMAP in the cold season. Snow, heavy canopy, and land development cause SMAP retrievals to have errors. In addition, this region does not have a good coverage of soil moisture and has less number of retrievals per grid (Figure 1). However, the KS filter complements the map by showing that the long-term and short-term VIC during cold season stay pretty stationary over time. This means that the soil moisture in this area has been less subject to change during cold season at least for the past 40 years.

This information can be used to inform an interpretation of SMAP soil moisture percentiles maps based on $< 10$ years of data, as presented in Figures 7 and 8 for a selection of soil moisture drought and flood indices. The grids that fail both KS and correlation tests (white grids in Figure 6) will be flagged and are where we have the highest uncertainty of the quality of the data. This includes about 500 grids in the warm season and about 350 grids in cold season over the CONUS.

### 3.3 Comparison of the drought indices

In Figure 7 to Figure 10, several indices are compared to the SMAP-based drought index. For surface soil moisture index based on SPL3SMP, we provide a 3-day composite SMAP index to offer more continuous coverage. The union filter is applied to omit the grids that do not have reliable estimates. Our index SPL3SMP index maps are compared with the 1-month SPI (SPI-1)

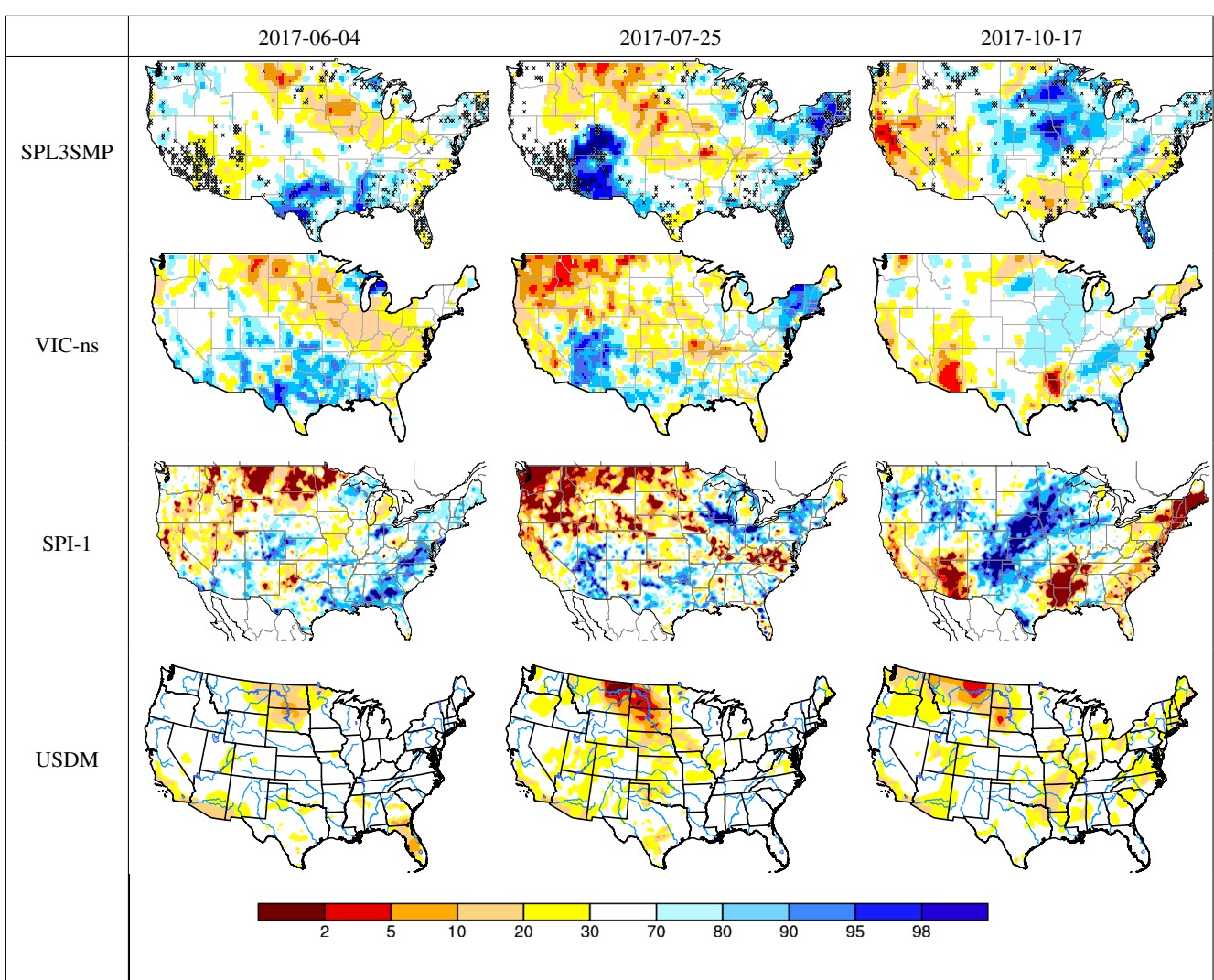

**Figure 7.** Comparison between SPL3SMP index map and VIC-ns, SPI-1, and USDM in 2017. The black x symbols in the SPL3SMP maps are the grids that passed neither filter and were shown as white grids in Figure 6. For USDM, drought levels from 30 to 100 are shown in white.

index, a VIC-ns index, and the USDM. For SMAP soil moisture index based on the SPL4SMAU, comparisons are made with a 3-month SPI (SPI-3) index and a GRACE satellite product. All the products except for GRACE were described in Section 1. GRACE is NASA's Gravity Recovery and Climate Experiment (GRACE) satellite system that detects small changes in the Earth's gravity field caused by the redistribution of water on and beneath the land surface. Combined with the Catchment Land

5 Surface Model using an Ensemble Kalman smoother data assimilation Zaitchik et al. (2008), GRACE maps root zone soil moisture and groundwater transformed into percentiles (NDMC, 2018b).

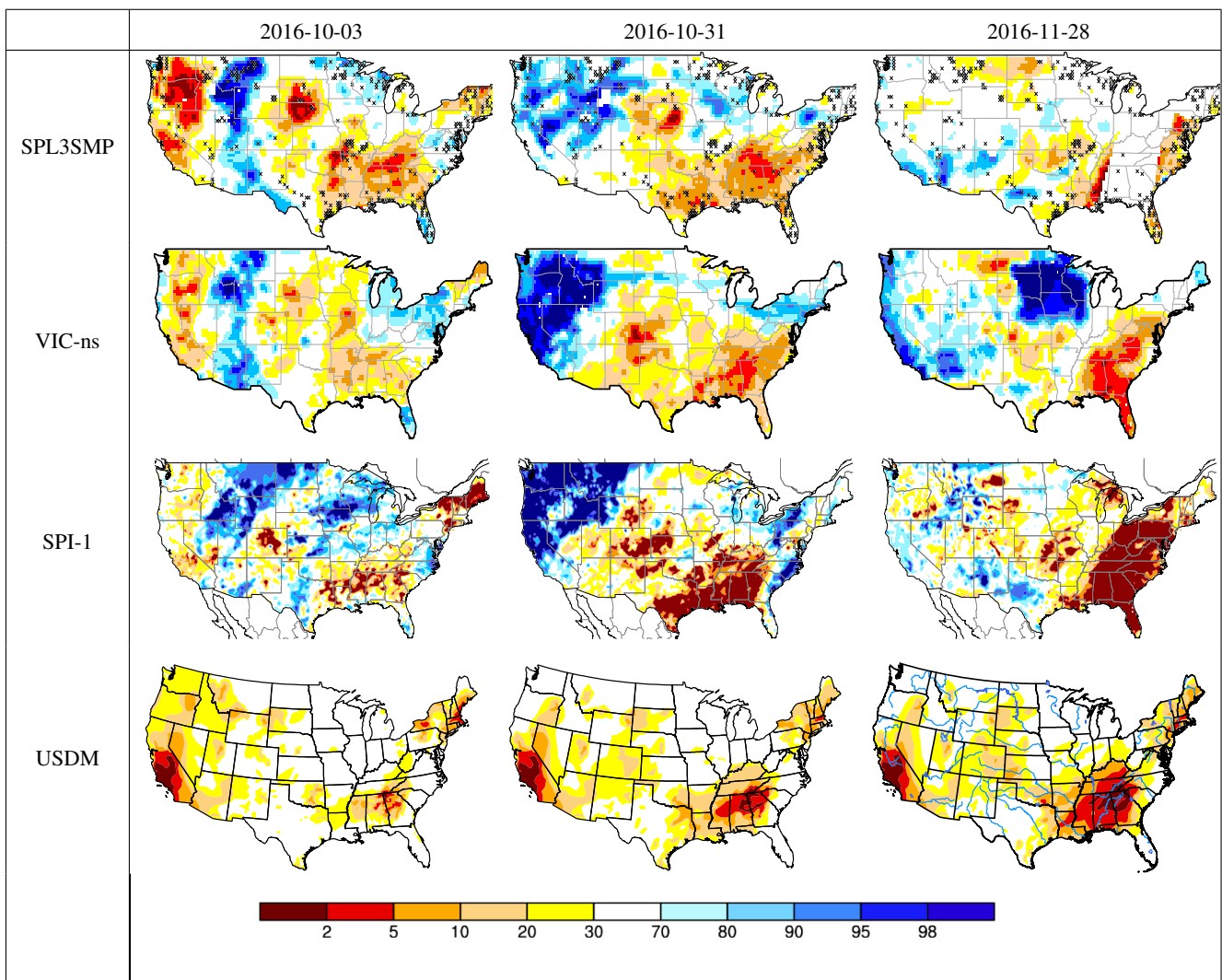

**Figure 8.** Comparison between SPL3SMP index map and VIC-ns, SPI-1, and USDM in 2016. The black x symbols in the SPL3SMP maps are the grids that passed neither filter and were shown as white grids in Figure 6. For USDM, drought levels from 30 to 100 are shown in white.

Figure 7 and Figure 9 show drought during the period from June 4 through October 17, 2017, for both near surface and root zone. In this period, there was one agricultural drought event in Montana, and North and South Dakota, with losses exceeding $1 billion across the United States (NOAA, 2018b). The plains of eastern Montana experienced exceptional drought throughout July to October 2017 and in late October drought started to recover. The peak of the drought was in July 2017 when 20% of Montana was in severe drought and 23% of it in moderate drought. Concurrently, 40% of North Dakota was in extreme drought while 70% of the state was under some level of drought, and similarly, 68% of South Dakota was under severe drought (NOAA, 2018b). Both SPL3SMP and SPL4SMAU index maps seem to catch this drought event, although the event was more

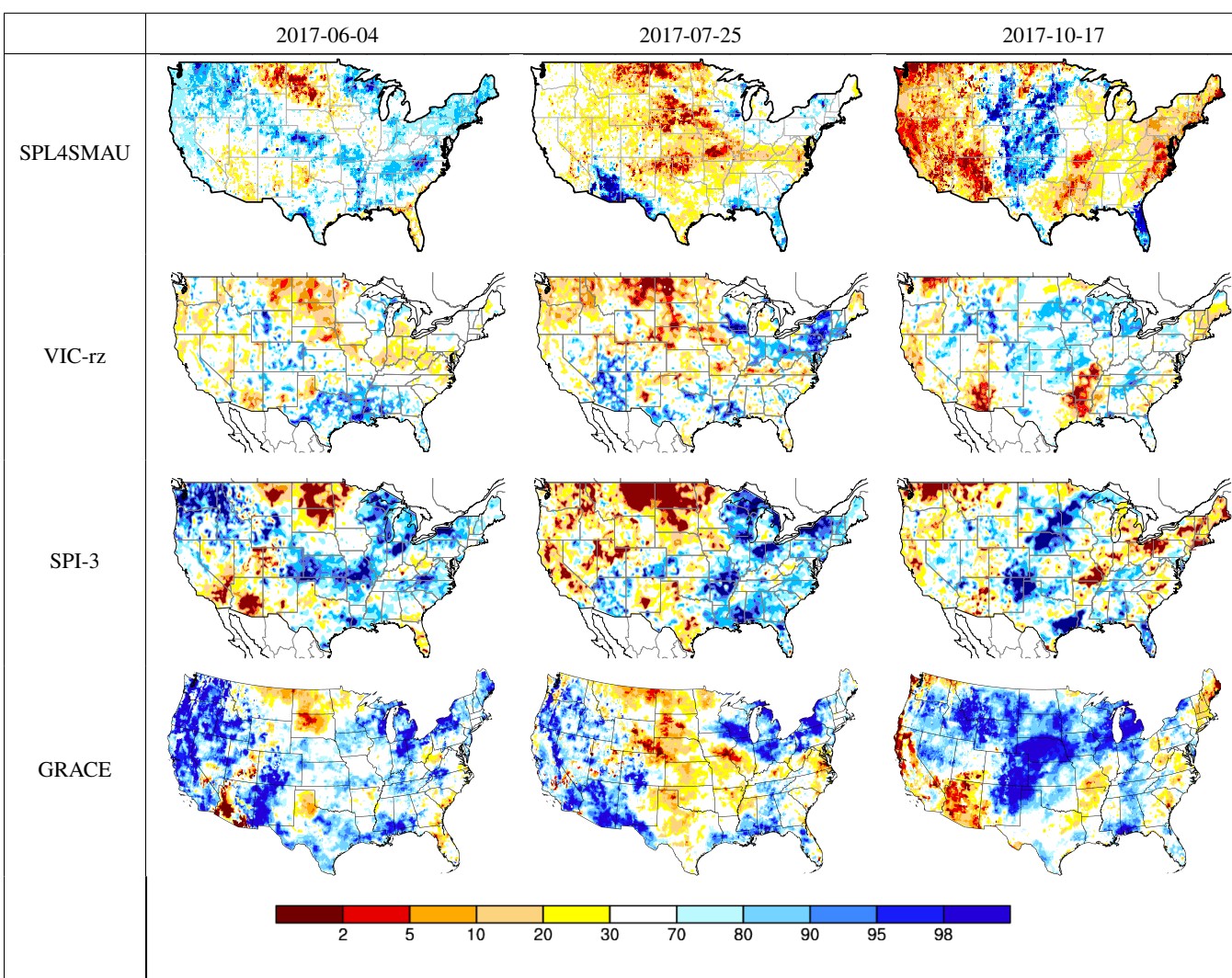

|  | 2017-06-04 | 2017-07-25 | 2017-10-17 |
|---|---|---|---|
| SPL4SMAU | | | |
| VIC-rz | | | |
| SPI-3 | | | |
| GRACE | | | |

**Figure 9.** Comparison between SPL4SMAU index map and VIC-rz, SPI-3, and GRACE in 2017.

pronounced in the root zone than the surface. The maps of these two figures are also in general agreement. It is important to clarify that for 2017 period, the GRACE sensor was failing and the resulting water storage observations were unreliable. Therefore, the last GRACE gravity field retrievals processing only go through June 2017. Therefore, GRACE NDMC results associated with Figure 9 are not consistent with other products and likely do not reflect actual GRACE observations for 2017.

5    In Figure 8 and Figure 10, drought during the period of October 3 to November 8, 2016 is shown for both near surface and root zone. In 2016, there were three drought events in the western, northeastern and southeastern parts of the U.S. which are captured by both SPL3SMP and SPL4SMAU index maps. The drought had mostly been alleviated in northern California by near-normal precipitation during the 2015-16 Winter, and above normal precipitation in Fall 2016. To the extent that the

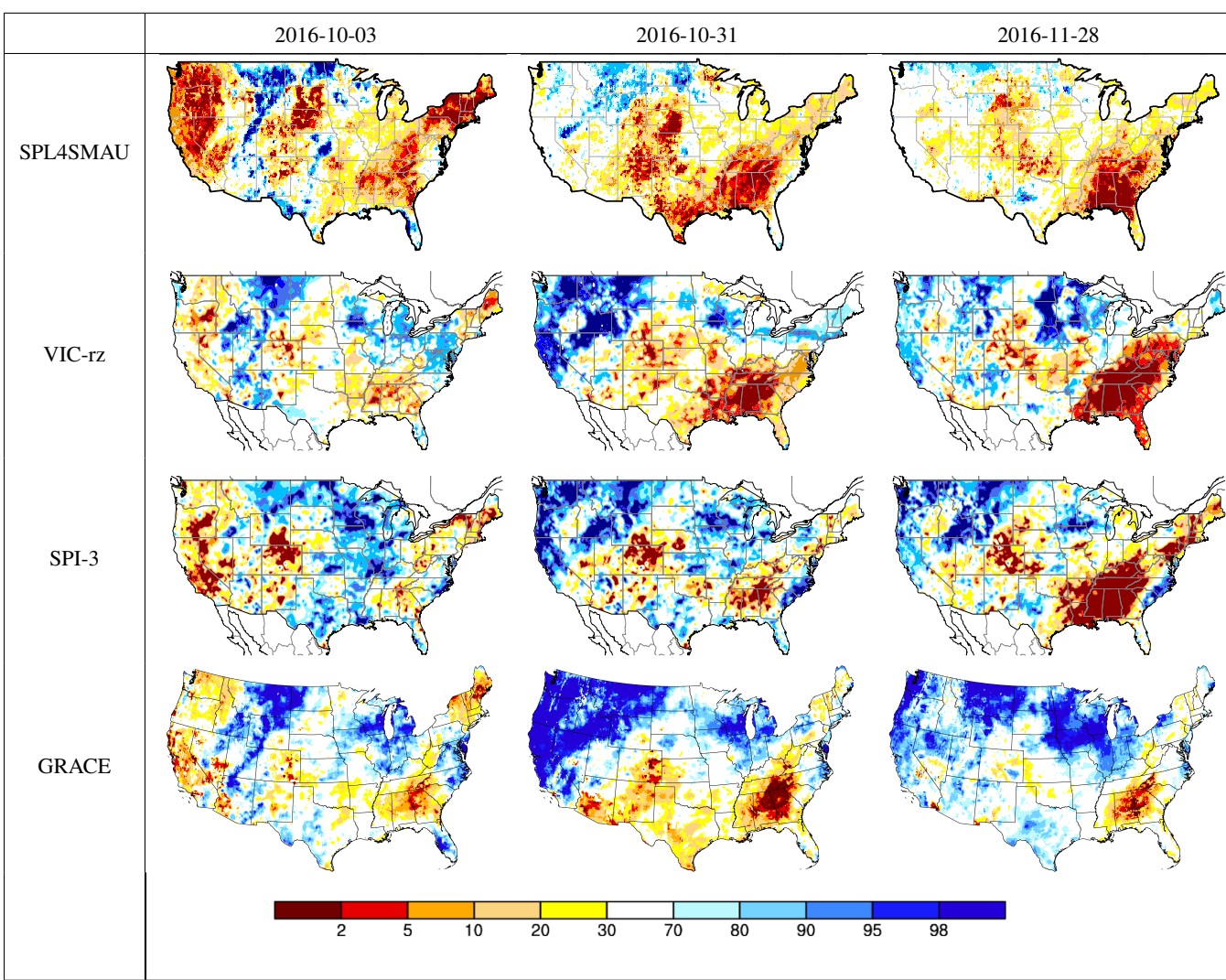

**Figure 10.** Comparison between SPL4SMAU index map and VIC-rz, SPI-3, and GRACE in 2016.

drought persisted in Southern California after this period, it is reflected in total column soil moisture rather than near-surface soil moisture (Figure 9).

There is a high correspondence among the drought maps, particularly in the development of the drought in the southeastern U.S. during October and November 2016. Due to heavy rainfall along the Mississippi River in November, the drought migrated
5   eastwards. Also, by November 2016 the drought in southern California was alleviated, which is picked up by SPL3SMP, SPL4SMAU, VIC-ns and VIC-rz, SP-1 and 3, GRACE, and to a much lesser extent by the USDM that showed an increasing area under drought on November 28 compared to SPL3SMP, SPL4SMAU, GRACE, or VIC-ns and VIC-rz. Additionally, for the maps that also include wetness (all except USDM), there is a high correspondence of pluvial regions (example Figure 7).

Most of the grids where we do not have confidence in the accuracy of predictions are in Southern California and Nevada during the warm season (eg. SPL3SMP index map on 2017-06-04 and 2017-07-25 in Figure 7). In fact, there is a visible discrepancy between SPL3SMP and VIC-ns index maps during that period in Southern California. We believe this is due to lack of correlation between SPL3SMP and VIC-ns in that area since VIC does not model regulation. Human interference and use of groundwater wells during warm season can play a major part in what VIC models and what SMAP sees. For that reason, we think SMAP's metrics in the area are more accurate than from VIC-ns.

## 4    Conclusions

The drought index described in this study provides a reliable estimate of the state of drought on a daily basis for the CONUS using SMAP. We fitted beta distributions to the SMAP data and used correlation, KS, and a combination of those two filters to numerically assess the adequacy of the short-term SMAP data for each grid cell. The areas that passed neither the KS nor correlation tests were flagged in the final SMAP drought index. These areas are grids where we have less confidence in reliable drought index estimates: they are non-stationary and thus their soil moisture has been changing over the past 40 years. The flagged grids can be seen as an adjustment to the model to remove non-climatic influences or water management practices, although more in-depth research is needed to confirm such changes. Given the limited scope of the data, the results should be considered a demonstration of the reliability and usefulness of SMAP for a drought monitoring product and for implementation into an operational drought-monitoring tool.

Besides drought, SMAP can also identify regions of anomalously wet conditions that can be of great use to water and agricultural managers. Wet indices can indicate potential flood-prone conditions and therefore regions can be put on flood alerts if additional heavy rain occurs. Also, wet conditions can impact farm management, especially in the spring when sowing takes place or during the harvesting period.

Through comparing SMAP based index maps for drought and wet conditions with other index products we see a high similarity. Although there can be some errors at different levels, the overall evaluation reveals that SMAP based drought products can be a viable alternative for drought monitoring in the U.S. This is advantageous since SMAP is generated at a daily resolution with almost complete coverage every three days. This enables observing the effect of fluctuations in other hydrological variables, such as precipitation. In comparison, USDM, GRACE, and SPI have a low temporal resolution which makes it difficult to study the shorter-term impacts from the other variables on soil moisture.

Both near surface and root zone soil moisture drought products can provide important information about the availability of soil moisture at the stage where plants develop in order to cultivate the optimum harvest. Future applications of this study can be coupling plant growth models with near surface and root zone soil moisture drought index products (NDMC, 2018a).

The soil moisture data are a culmination of all hydrological processes and represent available water from incoming precipitation and throughfall to evapotranspiration and drainage processes. The SMAP satellite is providing global observations of soil moisture of unprecedented quality. Because SMAP monitors soil moisture directly and provides critical information for drought early warning, it is important that the future developments focus on drought assessment using SMAP in underrep-

resented parts of the world. Thus results here provide significant support for a global SMAP drought and pluvial conditions monitoring system. since SMAP data can be retrieved and maps can be generated in near-real time, it is very promising that a SMAP drought index product can be implemented operationally.

*Data availability.* The SMAP-based drought index product at the daily resolution for CONUS is available at http://hydrology.princeton.edu/
5   smap-drought

*Acknowledgements.* This work was supported by NASA grant CNV1003235. The USDM and GRACE maps were provided by the National Drought Mitigation Center at the University of Nebraska-Lincoln and were downloaded from their websites at http://droughtmonitor.unl.edu and athttp://nasagrace.unl.edu, respectively. The VIC data were provided by Princeton University's Terrestrial Hydrology Group. This paper
10  benefited greatly from the reviewers' comments. We thank them for their time and support.

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
