# Peer review of "Developing a drought monitoring index for the Contiguous U.S. using SMAP"

_Hydrology and Earth System Sciences, 2018_

## Referee Comment (RC1) · Anonymous Referee #1 · 18 May 2018

With interest I have read the manuscript entitled "A SMAP-Based Drought Monitoring Index for the United States", it is interesting and well written (although I am not a native English speaker!). It is certainly of interest for HESS readers. The manuscript details a new SMAP-based index for drought monitoring over the Continental US (although the title mentions " [...] the United States). The methodology stems on previous work from Sheffield et al., 2004 and is applied to the recent SMAP data. Te resulting drought index is then compared to other already existing index like SPI-1&3, and another one GRACE-based. While the article is clear (at least to me), I am missing some more analyses of the drought Index for the manuscript to go from ' [...] a demonstration of the reliability [...]' with lot of text to a proper journal article (There is no results section?). Authors present several comparisons but not a proper evaluation of the added value

of this index. Some more in depth analyses of the added value of the new Index otherwise it is simply one Index amongst many other. For example, if you were using NASA's Catchment LSM (i.e., without assimilation of any SMAP data) would you have different results? How could you / would you quantify/highlight the added value o f SMAP?

Below are some suggestions/corrections that have to be accounted for before the manuscript deserves publication.

Abstract General comment : Some parts seem awkwardly written and are not self explanatory, not all acronyms are given and some very specifics information are given making it difficult to follow for the Reader. P.1,L.5: '[...] so the 33 months [...]', when doest it starts and when does it stops? P.1,L.6: Please clarify SPL4SMAU, while I assume it is the level 4 products it might not be obvious for everyone. P.1,L.10: if your intention is to say that your drought index is based on the level 4 product (SPL4SMAU), simply say it. P.1,L.12: it might not be obvious for everyone what is VIC, please clarify. P.1,L.13: '[...] 57% of grids[...]', please clarify. P.1,L.16-17: Not obvious what is D0-D4, GRACE and W0-W4, please clarify.

Introduction General comment: Somme paragraphs are a bit long and could be shortened (it is just my opinion so I let this comment to Author's discretion). P.2, L.8: what is NCEI? P.2, L.10: what is UNPD P.3, L.30: '[...] differs a great deal [...]', please consider rephrasing. P.4, L.5: '[...] and others [...]', please consider using 'et. al.' P.4, Is the last paragraph of the introduction in agreement with the four point described above (I am thinking of point 3 in particular). Maybe a link could be made with long data record of soil moisture from satellite derived surface soil moiture like the ESA-CCI data set (e.g., Dorigo et al., 2017, see reference below). P.4, L.18 : please clarify W0-W4 (I guess W is for week...) Dorigo, W., Wagner, W., Albergel, C. Albrecht, F., Balsamo, G., Brocca, L., Chung, D., Ertl, M., Forkel, M., Gruber, A., Haas, E., Hamer, P. D., Hirschi, M., Ikonen, J., de Jeu, R., Kidd, R., William Lahoz g, Liu, Y. Y., Miralles, D., Mistelbauer, T., Nicolai-Shaw, N., Parinussa, R., Pratola, C., Reimer, C., van der Schalie, R., Seneviratne, S. I., Smolander, T., and Lecomte, P.:ESA CCI soil moisture for improved Earth system understanding: state-of-the art and future directions, Remote Sens. Environ., 201, RSE-10331, https://doi.org/10.1016/j.rse.2017.07.001, 2017.

Data and Methods General comment : I believe consistency is a key element that could be improved. P.4, L.22: 'Since 31 March [...]', in the introduction it is since April. P.4, L.26-27: please rephrase to mention Level 3; Level 4... P.4, L.33: If it is he case (and I think it is from the introduction), the SMAP L-band Tb that are assimilated to produce the SPL4SMAU are the SPL3SMAU data (?) if so, please simply say it. P.5, L.9, what is 'L4', please clarify. P.5, L.27, '33 months' sometimes, '1009 days' some others, please be consistent if you are talking about the same thing. P.6, L.15, please rephrase question. P.6, L.29, what is 'mpas' ? Please correct typo ('maps' I guess) Figure 2: units? P.8, L.1: Please comment on correlation values and their significance.

Comparison to other indices General comment : I am missing some in depth analyses of the added value of the new Index otherwise it is simply one Index amongst many other. For example, if you were using NASA's Catchment LSM (i.e., without assimilation of any SMAP data) would you have different results? P.10, I think a word is missing in the second sentence (?) P.11, why are the figures embedded in the conclusion? '[...][...]'

Conclusions Some sentences really look like introduction to me.

---

## Referee Comment (RC2) · Anonymous Referee #2 · 25 Jul 2018

The manuscript describes the use of SMAP soil moisture data products to generate a percentile-based soil moisture product for drought/pluvial monitoring. There is some interesting and highly relevant material here; however, as currently written, the manuscript lacks any firm conclusions and/or meaningful analysis. As a result, it reads more like a technical/progress report than an actual journal paper.

MAJOR

This shortcoming can be fixed by providing a more direct link between the (very interesting) "data adequacy" analysis presented in Section 2.3 and the presentation of index comparisons in Section 3. As currently written, the analysis in Section 2.3 reveals that the (current) 3-year SMAP data heritage is insufficient for a substantial fraction of CONUS. However, this "inadequacy" is never mentioned again in the paper and does

not come into the analysis of results presented in Section 3 and discussed in Section 4. This is a real shame. At best, SMAP will last for 10 years; therefore, "data adequacy" will always be a pressing concern for the calculation of soil moisture climate percentiles.

Given this pressing need - how can the analysis in Section 3 be used to inform an interpretation of SMAP soil moisture percentile maps based on <10 years of data (e.g., as a tool for generating data quality flags, as a data mask or as a source of uncertainty information)? Does the fit between these new SMAP-based indices and existing drought/pluvial indices noticeably degrade for areas flagged as "inadequate" in Figure 5? Are there specific events there where the 3-year SMAP data record injects spurious percentile patterns into drought/pluvial events? If so, are the locations of these events adequately flagged as being problematic by results in Figure 5?

More analysis on these (and related) questions would greatly enhance the contribution of the manuscript (which currently is somewhat poorly defined).

MODERATE

1) Figure 2 – A major issue is calculating percentile products is always determining the seasonal intervals over which climate is considered stationary. Here, the authors choose to (implicitly) assume stationary climate within "hot" and "cold" 6-month portions of the year. Some discussion supporting this choice would be helpful. For instance, the warm versus cold season soil moisture differences in Figure 2 are (surprisingly) quite small. On the face of it, this lack of seasonality probably supports the author's decision to consider seasonality in a relatively simple way.

2) Page 8/Lines 4-7 – The attribution of this "Southern California" signal to an irrigation effects is problematic. The area fraction of Southern California that is irrigated is actually quite low. It is much more likely that the lack of (VIC/SMAP) correlation in these areas is due to thermal problems with 6 pm retrievals over arid/semi-arid regions (which is why the problem does not re-occur in Nebraska) during the summer (basically, summertime pm conditions violate the soil/canopy isothermal assumption that SMAP uses to retrieve soil surface moisture). One way to test this, would be to re-generate Figure 4a using only 6 am retrievals and see if the effect goes away.

3) Bottom of page 8...what exactly is meant by "raw" SMAP retrievals? Also, the list here seems to contain 4 comparisons not 6 (as stated in the text). Finally, the exact link between these 4 (or 6) comparisons and plotted results in Figure 3 is a bit unclear. A couple more explanatory sentences would help here.

4) Bottom of page 13/of page 14. It is not clear to me how the SMAP L4 product could possibly detect the impact of groundwater extraction (using a land model which does not consider the impact of well pumping on saturated zone calculations and assimilation observations sensitive to only the top 5 cm of the soil column). Therefore, the attribution presented here seems potentially misguided. This discussion should be either strengthened or removed.

Minor notes:

1) The abstract spends too much time discussing SMAP background (in the first paragraph) and too little time defining the contribution of this particular manuscript (see major point above).

2) The SMAP product version names in the manuscript differ from the "official" product names/acronyms (see https://smap.jpl.nasa.gov/data/)...good to use the official versions.

3) Page 3/Line 20...double parentheses.

4) Page 7/Line 4...better to say "too tightly bounded".

5) Page 7/Lines 9-11...reword to clarify...unclear how the moment matching approach applied here differences from that of Sheffield et al. (2004).

6) Figure 5 needs a color key...not clear what grey shading indicates.

7) Bottom of page 12...where exactly is this "grid analysis" presented? Unclear what is being referred to here.

---

## Author Comment (AC1) · 10 Sep 2018

**Response to Referee #1**

Paper's New Title: Developing a drought monitoring index for the Contiguous U.S. using SMAP

September 10, 2018

1. Comments from referee
   With interest I have read the manuscript entitled "A SMAP-Based Drought Monitoring Index for the United States", it is interesting and well written (although I am not a native English speaker!). It is certainly of interest for HESS readers. The manuscript details a new SMAP-based index for drought monitoring over the Continental US (although the title mentions [...] the United States).

2. Author's response
   Thank you for this comment. We have changed the title to the one above. We believe is targeting the content of the paper.

3. Author's changes in manuscript
   Paper's New Title: Developing a drought monitoring index for the Contiguous U.S. using SMAP

1. Comments from referee
   The methodology stems on previous work from Sheffield et al., 2004 and is applied to the recent SMAP data. The resulting drought index is then compared to other already existing index like SPI-1&3, and another one GRACE-based. While the article is clear (at least to me), I am missing some more analyses of the drought Index for the manuscript to go from [...] a demonstration of the reliability [...] with lot of text to a proper journal article (There is no results section?).

2. Author's response
   There are some differences between our approach and those from Sheffield et al., 2004. After carefully revisiting there paper, we made major numerical analysis on the added value of the drought index using SMAP and expanded our Results section. To assess the data adequacy and have confidence in using short-term SMAP for drought index estimate, we analyzed individual grids by defining two filters and a combination of them, which could separate 5,815 grids covering CONUS into passed and failed grids. The two filters were: (1) The Kolmogorov-Smirnov (KS) test for beta-fitted long-term and short-term Variable Infiltration Capacity (VIC) LSM with 95% confidence; and (2) Good correlation (0.4) between beta-fitted VIC and beta-fitted SPL3SMP. To evaluate which filter is the best, we defined a Mean Distance ($MD$) metric, assuming VIC index at 36 km resolution is the ground truth. The new reliability analysis is described under Data adequacy filter section.

3. Author's changes in manuscript

[revised manuscript text omitted]

1. Comments from referee
   Authors present several comparisons but not a proper evaluation of the added value of this index. Some more in depth analyses of the added value of the new Index otherwise it is simply one Index amongst many other. For example, if you were using NASAs Catchment LSM (i.e., without assimilation of any SMAP data) would you have different results? How could you / would you quantify/highlight the added value of SMAP?

2. Author's response
   We agreed that some more in-depth analysis of the usefulness of the index would have added more value to the process and the paper. Hence we did extensive analysis on the reliability of our work by analyzing individual grids by defining two filters and a combination of them, which could separate 5,815 grids covering CONUS into passed and failed grids. The two filters were: (1) The Kolmogorov-Smirnov (KS) test for beta-fitted long-term and short-term VIC with 95% confidence; and (2) Good correlation ($> 0.4$) between beta-fitted VIC and beta-fitted SPL3SMP. The process is described in the previous response. Below we should the numerical analysis results. The index without NASA's LSM looks like Level 3 data.

3. Author's changes in manuscript

[revised manuscript text omitted]

1. Comments from referee
   Abstract General comment : Some parts seem awkwardly written and are not self explanatory, not all acronyms are given and some very specifics information are given making it difficult to follow for the Reader.

2. Author's response
   In addition to carefully addressing your specific comments about all the acronyms, we re-read and

reevaluated the abstract and rewrote it to make sure it is written in more fluent and scientific way. Please see the new version.

3. Author's changes in manuscript

Abstract: Since April 2015, NASA's Soil Moisture Active Passive (SMAP) mission has monitored near-surface soil moisture, mapping the globe (between $85.044°N/S$) using an L-band (1.4 GHz) microwave radiometer in 2-3 days depending on location. Of particular interest to SMAP-based agricultural applications is a monitoring product that assesses the SMAP near-surface soil moisture in terms of probability percentiles for dry and wet conditions. However, the short SMAP record length poses a statistical challenge for meaningful assessment of its indices. This study presents initial insights about using SMAP for monitoring drought and pluvial regions with a first application over the Contiguous United States (CONUS). SMAP soil moisture data from April 2015 to December 2017 at both near-surface (5cm) SPL3SMP, or Level 3, at ∼36 km resolution; and root zone SPL4SMAU, or Level 4, at ∼9 km resolution were fitted to beta distributions and were used to construct probability distributions for warm (May-October) and cold (November-April) seasons. To assess the data adequacy and have confidence in using short-term SMAP for drought index estimate, we analyzed individual grids by defining two filters and a combination of them, which could separate 5,815 grids covering CONUS into passed and failed grids. The two filters were: (1) The Kolmogorov-Smirnov (KS) test for beta-fitted long-term and short-term Variable Infiltration Capacity (VIC) LSM with 95% confidence; and (2) Good correlation ($\geq 0.4$) between beta-fitted VIC and beta-fitted SPL3SMP. To evaluate which filter is the best, we defined a Mean Distance ($MD$) metric, assuming VIC index at 36 km resolution is the ground truth. For both warm and cold seasons, the union of the filters – which also gives the best coverage of the grids throughout CONUS – was chosen to be the most reliable filter. We visually compared our SMAP-based drought index maps with metrics such as U.S. Drought Monitor (from D0-D4), SPI 1 month and VIC near surface from Princeton University. The root zone drought index maps were shown to be similar to those produced by the VIC at root zone, SPI 3 month, and GRACE. This study is a step forward towards building a national and international soil moisture monitoring system, without which, quantitative measures of drought and pluvial conditions will remain difficult to judge.

1. Comments from referee
P.1,L.5: [...] so the 33 months [...], when doesn't it starts and when does it stops?

2. Author's response
Since we rewrote the abstract this sentence has been changed.

3. Author's changes in manuscript
Please read the new abstract in previous response.

1. Comments from referee
P.1,L.6: Please clarify SPL4SMAU, while I assume it is the level 4 products it might not be obvious for everyone.

2. Author's response
Since we rewrote the abstract this sentence has been changed.

3. Author's changes in manuscript
The sentence changed to: SMAP soil moisture data from April 2015 to December 2017 at both near-surface (5cm) SPL3SMP ( 36km res.) and root zone SPL4SMAU ( 9km res.) were fitted to a beta distribution and were used to construct probability distributions for warm (May-October) and cold (November-April) seasons.

1. Comments from referee
P.1,L.10: if your intention is to say that your drought index is based on the level 4 product (SPL4SMAU), simply say it.

2. Author's response
Since we rewrote the abstract this sentence has been changed here. Regardless, our drought indices are both based on Level 3 and Level 4.

3. Author's changes in manuscript
Since we changed the previous sentence to address the issue, this sentence was removed.

1. Comments from referee
P.1,L.13: [...] 57% of grids[...], please clarify.

2. Author's response
The abstract is rewritten to fit the new numerical analysis and hence, doesnt include this part any longer.

3. Author's changes in manuscript
The 57% is clarified more in caption of Figure 2. Please see Figure 2.

1. Comments from referee
P.1,L.16-17: Not obvious what is D0-D4, GRACE and W0-W4, please clarify. Introduction General comment: Some paragraphs are a bit long and could be shortened (it is just my opinion so I let this comment to Authors discretion).

2. Author's response
The sentence changed to: W0-W4 was removed.

3. Author's changes in manuscript
NA

1. Comments from referee
P.2, L.8: what is NCEI?

2. Author's response
This was addressed and changed to NOAA. In the references more information is given.

3. Author's changes in manuscript
Example: In the U.S. since 1996, there has been at least one drought event per year except for years 1997, 2001, 2004, and 2010, and each year drought cost between 1 billion and 14 billion dollars in damages (in 2015 - adjusted dollars) (NOAA, 2018).

1. Comments from referee
P.2, L.10: what is UNPD

2. Author's response
Sorry for the typo. It was UNDP and regardless are more recent UN-ISDR article was more relevant. We referenced UN-ISDR instead.

3. Author's changes in manuscript
Although the impacts of drought are intimately linked to the vulnerability of a population to adverse conditions (UN/ISDR, 2007) and how society responds within the constraints of changing economies, timely determination of the current level of agricultural drought aids the decision-making process in order to reduce its impacts.

1. Comments from referee
P.3, L30: [...] differs a great deal [...], please consider rephrasing.

2. Author's response
Reworded to differs considerably.

3. Author's changes in manuscript
Additionally, intercomparison of the four NLDAS models showed that soil moisture differs considerably among models (Robock et al., 2000).

1. Comments from referee
   P.4, L.5: [...] and others [...], please consider using et. al.

2. Author's response
   This is fixed.

3. Author's changes in manuscript
   An alternative approach to using model-derived soil moisture for drought detection and prediction is satellite-derived soil moisture. There are currently four major satellite-based systems that provide soil moisture products at various spatial and temporal resolutions: MetOp with the advanced scatterometer (ASCAT) (Brocca et al., 2010; Wagner et al., 2013), JAXA's Advanced Microwave Scanning Radiometer AMSR2 (Parinussa et al., 2015; Wu et al., 2015) with the C- and X Band passive radiometers on the GCOM-W1 satellite that is a follow-on to the AMSR-E sensor, which failed on 4 October 2011 and was part of NASA's Earth Observing System; ESA's Soil Moisture Ocean Salinity (SMOS) L-band radiometer (Pan et al., 2010; Kerr et al., 2012, 2016) and NASA's Soil Moisture Active Passive (SMAP) L-band radiometer Entekhabi et al. (2010).

1. Comments from referee
   P.4, Is the last paragraph of the introduction in agreement with the four point described above (I am thinking of point 3 in particular). Maybe a link could be made with long data record of soil moisture from satellite derived surface soil moisture like the ESA-CCI data set (e.g., Dorigo et al., 2017, see reference below).

2. Author's response
   Shortness of data availability for SMAP is a challenge, However, we numerically figured out how to address this challenge. We looked at the correlation between SMAP and VIC and distribution parameters between long term VIC and short term VIC and could make reasonable conclusion on how to overcome the challenge and how reliable our estimates of the drought index using SMAP can be. We wrote about these approaches in the Data adequacy, Result, and Discussion sections. We have decided not to include the ESA-CCI in the analysis.

3. Author's changes in manuscript
   Please refer to above methodology explanations.

1. Comments from referee
   P.4, L.18 : please clarify W0-W4 (I guess W is for week)

2. Author's response
   We removed this expression from the paper. They refer to the drought levels in the USDM. We don't know what W stands for (perhaps wet?).

3. Author's changes in manuscript
   NA

1. Comments from referee
   Data and Methods General comment : I believe consistency is a key element that could be improved.

2. Author's response
   We also appreciate noticing that and we have worked through the text and made sure all the dates are consistent from April 1, 2015 through December 31, 2017.

3. Author's changes in manuscript
   Various locations

1. Comments from referee
   P.4, L.22: Since 31 March [...], in the introduction it is since April.

2. Author's response
   This is fixed and all through the paper April 1st is indicated as the first day of SMAP analysis.

3. Author's changes in manuscript
Since April 2015, NASA's SMAP mission has been monitoring near-surface soil moisture, mapping the globe (between $85.044°N/S$) using an L-band (1.4 GHz) microwave radiometer in 2-3 days depending on location.

1. Comments from referee
P.4, L.26-27: please rephrase to mention Level 3; Level 4...

2. Author's response
Fixed!

3. Author's changes in manuscript
The SMAP mission provides a set of operational global data products that include:

  – Level 3 (SPL3SMP): a composite based on daily passive radiometer estimates of global land surface soil moisture (nominally 5 cm) that are resampled to a global, cylindrical 36 km Equal-Area Scalable Earth Grid, Version 2.0 (EASE-Grid 2.0) (O'Neill et al., 2016). Regions of heavy vegetation (vegetation water content $> 4.5\ kg/m^2$) or frozen ground or snow covered are masked out using a Normalized Polarization Ratio (NPR)-based passive freeze-thaw retrieval. Given the 1000-km swath and 98.5 minute orbit, the SPL3SMP retrievals are spatially and temporally discontinuous with 2-3 day gaps depending on location; and

  – Level 4 (SPL4SMAU): provides global surface and root zone soil moisture by assimilating the SMAP L-band brightness temperature data (for which SPL3SMP is the gridded version) from descending and ascending half-orbit satellite passes, approximately 6:00 a.m. to 6:00 p.m., every 3 hours, local solar time, into NASAs Catchment LSM (Reichle, 2017; Reichle et al., 2015). The SPL4SMAU data product is gridded using an Earth-fixed, global, cylindrical 9 km EASE-Grid 2.0 projection. The land surface model component of the assimilation system is driven by a forcing data stream from the global atmospheric analysis system at the NASA GMAO (Rienecker and coauthors, 2008). Additional corrections are applied using gauge- and satellite-based estimates of precipitation that are downscaled to the temporal and 9 km scale of the model forcing using the disaggregation methods described in Liu et al. (2011) and Reichle et al. (2011). The SPL4SMAU product provides global soil estimates for the surface (0-5 cm) and "root zone" (0-100 cm), and is an effort to provide continuous, daily information without the discontinuous data provided by the SPL3SMP radiometer retrievals. Nonetheless, the only product that doesn't use ancillary meteorological data is the SPL3SMP soil moisture retrievals.

1. Comments from referee
P.4, L.33: If it is he case (and I think it is from the introduction), the SMAP L-band Tb that are assimilated to produce the SPL4SMAU are the SPL3SMAU data (?) if so, please simply say it.

2. Author's response
This paragraph is changed to the one brought up in the previous response.

3. Author's changes in manuscript
Level 4 (SPL4SMAU) provides global surface and root zone soil moisture by assimilating the SMAP L-band brightness temperature data (for which SPL3SMP is the gridded version) from descending and ascending half-orbit satellite passes, approximately 6:00 a.m. to 6:00 p.m., every 3 hours, local solar time, into NASAs Catchment LSM (Reichle, 2017; Reichle et al., 2015).

1. Comments from referee
P.5,L.9, what is L4, please clarify.

2. Author's response
L4 is SPL4SMAU. I changed it to be consistent with the rest of the text.

3. Author's changes in manuscript
This part of text is changed. Additionally I checked that everywhere in the text is consistent and is using either Level 4 or SPL4SMAU.

1. Comments from referee
   P.5, L.27, 33 months sometimes, 1009 days some others, please be consistent if you are talking about the same thing.

2. Author's response
   This was clarified and now it is consistent throughout the text as 1,006 days (not 1,009).

3. Author's changes in manuscript
   Our SMAP data records are from 2015-04-01 to 2017-12-31, which is equivalent to 1,006 days.

1. Comments from referee
   P.6, L.15, please rephrase question.

2. Author's response
   It was rephrased.

3. Author's changes in manuscript
   A main challenge is to fit the four parameters of beta distribution, given a set of empirical observations.

1. Comments from referee
   P.6, L.29, what is mpas ? Please correct typo (maps I guess)

2. Author's response
   Yes, fixed!

3. Author's changes in manuscript
   NA

1. Comments from referee
   Figure 2: units?

2. Author's response
   The unit is $m^3/m^3$ and it is added to the text.

3. Author's changes in manuscript
   top row: SMAP index for the warm season during summer for SPL3SMP top 5 cm soil moisture (a), 20th percentile; (b), average soil moisture; (c), 80th percentile; bottom row: as the top row but for the cold season. Total period is from 2015/04/01 to 2017/12/31. The soil moisture unit is $m^3/m^3$.

1. Comments from referee
   P.8, L.1: Please comment on correlation values and their significance.

2. Author's response
   The significance of the correlation values between VIC and SPL3SMP is that is allows us to use VIC as a tool for fitting beta distribution to SPL3SMP. Under section titled Correlation Filter we explained the significance of it in the revised version.

3. Author's changes in manuscript
   Please refer to the section "correlation filter" above.

1. Comments from referee
   Comparison to other indices General comment : I am missing some in depth analyses of the added value of the new Index otherwise it is simply one Index amongst many other. For example, if you were using NASA's Catchment LSM (i.e., without assimilation of any SMAP data) would you have different results? P.10, I think a word is missing in the second sentence (?)

2. Author's response
   This comment of the reviewer was brought up on the first page. Please see the repeated version above. In addition, in the revised version we added very interesting and in-depth numerical analysis, introducing the Mean Distance metrics and filters to find the most reliable grids for SMAP-based drought predictions, which adds greatly to the value of research. The development of a drought index

using SMAP has not been done before. Although other drought indices exists, they are model based being forced by precipitation. SMAP drought index is unique to be an index solely based on remotely sensed satellite data.

3. Author's changes in manuscript

The drought index described in this study provides a reliable estimate of the state of drought on a daily basis for the CONUS using SMAP. We fitted beta distributions to the SMAP data and used correlation, KS, and a combination of those two filters to numerically assess the adequacy of the short term SMAP data for each grid cell. The areas that passed neither the KS nor correlation tests were flagged in the final SMAP drought index. These areas are grids where we have less confidence in reliable drought index estimates: the are non-stationary and thus their soil moisture has been changing over the past 40 years. The flagged grids can be seen as adjustment to the model to remove non-climatic influences or water management practices, although more in-depth research is needed to confirm such changes. Given the limited scope of the data, the results should be considered a demonstration of the reliability and usefulness of SMAP for a drought monitoring product and for implementation into an operational drought-monitoring tool.

Besides drought, SMAP can also identify regions of anomalously wet conditions that can be of great use to water and agricultural managers. Wet indices can indicate potential flood-prone conditions and therefore regions can be put on flood alerts if additional heavy rain occurs. Also, wet conditions can impact farm management, especially in the spring when sowing takes place or during the harvesting period.

Through comparing SMAP based index maps for drought and wet conditions with other index products we see high similarity. Although there can be some errors at different levels, the overall evaluation reveals that SMAP based drought products can be a viable alternative for drought monitoring in the U.S. This is advantageous since SMAP is generated at a daily resolution with almost complete coverage every three days. This enables observing the effect of fluctuations in other hydrological variables, such as precipitation. In comparison, USDM, GRACE, and SPI have low temporal resolution which makes it difficult to study the shorter-term impacts from the other variables on soil moisture.

Both near surface and root zone soil moisture drought products can provide important information about the availability of soil moisture at the stage where plants develop in order to cultivate the optimum harvest. Future applications of this study can be coupling plant growth models with near surface and root zone soil moisture drought index products (**?**).

The soil moisture data are a culmination of all hydrological processes and represent available water from incoming precipitation and throughfall to evapotranspiration and drainage processes. The SMAP satellite is providing global observations of soil moisture of unprecedented quality. Because SMAP monitors soil moisture directly, and provides critical information for drought early warning, it is important that the future developments focus on drought assessment using SMAP in underrepresented parts of the world. Thus results here provide significant support for a global SMAP drought and pluvial conditions monitoring system. since SMAP data can be retrieved and maps can be generated in near-real time, it is very promising that a SMAP drought index product can be implemented operationally.

1. Comments from referee

P.11, why are the figures embedded in the conclusion?

2. Author's response

Thank you for pointing that out. That was an error with how LATEX compiled. That is fixed now.

3. Author's changes in manuscript

Changes are made in the new manuscript.

1. Comments from referee

Conclusions Some sentences really look like introduction to me.

2. Author's response

That is fixed as well! Please see the revised version.

3. Author's changes in manuscript

[revised manuscript text omitted]

---

## Author Comment (AC2) · 10 Sep 2018

**Response to Referee #2**

Paper's New Title: Developing a drought monitoring index for the Contiguous U.S. using SMAP

September 10, 2018

1. Comments from referee
   This shortcoming can be fixed by providing a more direct link between the (very interesting) data adequacy analysis presented in Section 2.3 and the presentation of index comparisons in Section 3. As currently written, the analysis in Section 2.3 reveals that the (current) 3-year SMAP data heritage is insufficient for a substantial fraction of CONUS. However, this inadequacy is never mentioned again in the paper and does not come into the analysis of results presented in Section 3 and discussed in Section This is a real shame.

2. Author's response
   We appreciate this comment and agree that more numerical analysis on the adequacy of the SMAP data would have substantially enhanced the scientific merit of the paper. Therefore, we are addressing the issue through introducing two filters and a combination of them. We have done major changes to the paper and added specific sections for in-depth numerical analysis of the confidence of SMAP-based drought index maps and the adequecy of the data. Please see the section Data Adequacy Filters and beyond that for considering all the changes and explanations we provided in the newer version of the paper.

3. Author's changes in manuscript

[revised manuscript text omitted]

1. Comments from referee
   At best, SMAP will last for 10 years; therefore, data adequacy will always be a pressing concern for the calculation of soil moisture climate percentiles. Given this pressing need - how can the analysis in Section 3 be used to inform an interpretation of SMAP soil moisture percentile maps based on <10 years of data (e.g., as a tool for generating data quality flags, as a data mask or as a source of uncertainty information)?

2. Author's response
   From our numerical tests and results provided the information can be used to inform an interpretation of SMAP soil moisture percentiles maps based on <10 years of data, as presented in Figures 7 and 8 for a selection of soil moisture drought indices. The grids that fail both KS and correlation tests (white grids in Figure 6) will be flagged as crossed and are where we have the highest uncertainty of the quality of the data. This includes about 500 grids in the warm season and about 350 grids in cold season over the CONUS.

3. Author's changes in manuscript
   Please see the extensive analysis and maps provided in the previous section. In our new version we have created a mask to filter grids where we don't have enough certainty in SPL3SMP drough index.

1. Comments from referee
   Does the fit between these new SMAP-based indices and existing drought/pluvial indices noticeably degrade for areas flagged as inadequate in Figure 5? Are there specific events there where the 3-year SMAP data record injects spurious percentile patterns into drought/pluvial events? If so, are the locations of these events adequately flagged as being problematic by results in Figure 5?

2. Author's response
   For this question, we changed Figure 7 and 8 and showed the inadequate areas (based on our new results)

[Figure]

Figure 7: Comparison between SPL3SMP index map and VIC-ns, SPI-1, and USDM in 2017. For USDM, drought levels from 30 to 100 are shown in white.

omitted. The results show that the fit between new SMAP-based indices and existing drought/pluvial indices noticeably degrade for areas failing the filter (cross hatched).

3. Author's changes in manuscript

**1.3 Comparison among the drought indices**

In Figure 7 to Figure 10, several indices are compared to the SMAP-based drought index. For surface soil moisture index based on SPL3SMP, we provide a 3-day composite SMAP to offer index more continuous coverage. The union filter is applied to omit the grids that do not have reliable estimates. Our index SPL3SMP index produc maps are compared with the 1-month SPI (SPI-1) index, a VIC-ns index, and the USDM. For SMAP soil moisture index based on the SPL4SMAU, comparisons are made with a 3-month SPI (SPI-3) index and a GRACE satellite product. All the products except for GRACE were described in Introduction. GRACE is NASA's Gravity Recovery and Climate Experiment (GRACE) satellite system that detects small changes in the Earth's gravity field caused by the redistribution of water on and beneath the land surface. Combined with the Catchment Land Surface Model using an Ensemble Kalman smoother data assimilation Zaitchik et al. (2008), GRACE maps root zone soil moisture and groundwater transformed into percentiles (NDMC, 2018b).

[Figure]

Figure 8: Comparison between SPL3SMP index map and VIC-ns, SPI-1, and USDM in 2016. For USDM, drought levels from 30 to 100 are shown in white.

Figure 7 and Figure 9 show drought during the period from June 4 through October 17, 2017, for both near surface and root zone. In this period, there was one agricultural drought event in Montana, and North and South Dakota, with losses exceeding $1 billion across the United States (NOAA, 2018). The plains of eastern Montana experienced exceptional drought throughout July to October, 2017 and in late October drought started to recover. The peak of the drought was in July 2017 when 20% of Montana was in severe drought and 23% of it in moderate drought. Concurrently, 40% of North Dakota was in extreme drought while 70% of the state was under some level of drought, and similarly, 68% of South Dakota was under severe drought (NOAA, 2018). Both SPL3SMP and SPL4SMAU index maps seem to catch this drought event.

In Figure 8 and Figure 10, drought during the period of October 3 to November 8, 2016 is shown for both near surface and root zone. In 2016, there were three drought events in the western, northeastern and southeastern parts of the U.S. which are captured by both SPL3SMP and SPL4SMAU index maps. The drought had mostly been alleviated in northern California by near-normal precipitation during the 2015-16 Winter, and above normal precipitation in the Fall 2016. To the extent that the drought persisted in Southern California after this period, it is reflected in total column soil moisture rather than near-surface soil moisture (Figure 9).

There is a high correspondence among the drought maps, particularly in the development of the drought in the southeastern U.S. during October and November 2016. Due to heavy rainfall along the Mississippi

[Figure]

Figure 9: Comparison between SPL4SMAU index map and VIC-rz, SPI-3, and GRACE in 2017.

River in November, the drought migrated eastwards. Also, by November 2016 the drought in southern California was alleviated, which is picked up by SPL3SMP, SPL4SMAU, VIC-ns and VIC-rz, SP-1 and 3, GRACE, and to a much lesser extent by the USDM that showed an increasing area under drought on November 28 compared to SPL3SMP, SPL4SMAU, GRACE, or VIC-ns and VIC-rz. Additionally, for the maps that also include wetness (all except USDM), there is a high correspondence of pluvial regions (example Figure 7).

Most of grids where we do not have confidence in the accuracy of predictions are in Southern California and Nevada during the warm season (eg. SPL3SMP index map on 2017-06-04 and 2017-07-25 in Figure 7). In fact, there is visible discrepency between SPL3SMP and VIC-ns index maps during that period in Southern California. We believe this is due to lack of correlation between SPL3SMP and VIC-ns in that area since VIC does not model regulation. Human interference and use of groundwater wells during warm season can play a major part in what VIC models and what SMAP sees. For that reason, we think SMAP's metrics in the area are more accurate than from VIC-ns.

1. Comments from referee

Figure 2  A major issue is calculating percentile products is always determining the seasonal intervals over which climate is considered stationary. Here, the authors choose to (implicitly) assume stationary climate within hot and cold 6-month portions of the year. Some discussion supporting this choice would be helpful. For instance, the warm versus cold season soil moisture differences in Figure 2 are (surprisingly)

[Figure]

|  | 2016-10-03 | 2016-10-31 | 2016-11-28 |
|---|---|---|---|
| SPL4SMAU | | | |
| VIC-rz | | | |
| SPI-3 | | | |
| GRACE | | | |

Figure 10: Comparison between SPL4SMAU index map and VIC-rz, SPI-3, and GRACE in 2016.

quite small. On the face of it, this lack of seasonality probably supports the authors decision to consider seasonality in a relatively simple way.

2. Author's response

When we looked at the frequency distributions of soil moisture data at each grid, the data seemed to be dominated by either low soil moisture (summer time) or high soil moisture data (winter time). Further analysis showed these to be related to the warm and cold season periods. Therefore, to capture this inter-seasonal behavior in soil moisture, we divided the record into a warm season (April - September) and a cold season (October - March). Dividing the year into warm and cold seasons enabled us to track the soil moisture dynamics, and thus the probability distribution and index seasonally. Ideally, we would have divided it into monthly data but there are insufficient observations. Ideally we would have divided it into monthly data but there were insufficient observations to do that. This lead to our decision to divide the data into two seasons.

3. Author's changes in manuscript

We looked at the frequency distribution of soil moisture data at each grid. The data seemed to be dominated by low soil moisture in the summertime, and high soil moisture in the wintertime. Therefore, to capture this inter-seasonal behavior in soil moisture, we divided the record into a warm season (April - September) and a cold season (October - March). Dividing the year into warm and cold seasons enabled us to track the soil moisture dynamics, and thus the probability distribution and index seasonally. Ideally,

we would have divided it into monthly data but there are insufficient observations.

1. Comments from referee
   Page 8/Lines 4-7 The attribution of this Southern California signal to an irrigation effects is problematic. The area fraction of Southern California that is irrigated is actually quite low. It is much more likely that the lack of (VIC/SMAP) correlation in these areas is due to thermal problems with 6 pm retrievals over arid/semi-arid regions (which is why the problem does not re-occur in Nebraska) during the summer (basically, summertime pm conditions violate the soil/canopy isothermal assumption that SMAP uses to retrieve soil surface moisture). One way to test this, would be to re-generate Figure 4a using only 6 am retrievals and see if the effect goes away.

2. Author's response
   We are a little bit confused on this comment. We did not use a 6pm retrievals. We have used 6 am retrievals for any location. Regarding Southern California comment, we can recognize that there is an attribution to irrigation in southern and south central California and the high number of private and state owned groundwater wells (there are also as many as 2 million water wells in California that contribute to irregularity of groundwater and affecting the soil moisture. They range from hand-dug, shallow wells to carefully designed large-production wells drilled to great depths) which makes the area regulated and hence what SMAP sees is different what a LSM like VIC expects to be the case, however, we are unsure about other reasons for this. On the other hand, our new analysis and filter doesnt show a very strong confidence during Warm season on Southern California and we explained it in the text. More data is needed before we can recognize further attributions to low correlation between VIC and SMAP in that region. In the northeast and during winter we have the problems of ground covered by snow which doesnt result in good SMAP coverage and the low number of days and overpasses (presented in Figure ??) during winter in northeast can play a role in low amount of data and poor correlation during cold season. We added this content to the paper and provided references.

3. Author's changes in manuscript
   In one part: In this study, SPL3SMP products from the 6:00 a.m. retrievals and SPL4SMAU products from 6:00 a.m. retrievals, are used in the analysis of soil moisture drought index. Our SMAP data records are from 2015-04-01 to 2017-12-31, which is equivalent to 1,006 days.

   In another part: Figure 4 shows that the average correlation for both warm and cold seasons are high and around 0.6. During the warm season, the Central Valley and Southern California, Florida, northeastern U.S., and north of Wisconsin and Minnesota show poor correlation with VIC, around 0.2. The extent of this poor correlation increases during the cold season for northeastern U.S., Wisconsin and Minnesota. Snow season results in poor SMAP coverage during winter time in those areas. In addition, the low number of overpasses (presented in Figure 1) during winter in northeast can play a role in low amount of data and poor correlation during cold season. Contrary to the warm season, southern California shows a high a correlation with VIC during the cold season, around 0.9. We attribute this change in southern and south central California from cold season to warm season to irrigation that SMAP picks up, but VIC doesn't since the version used here doesn't have water management effects. Land use/land cover map shows that about one third of these areas are irrigated vegetation and another third is forests and woodlands (USGS, 2018). There are also as many as 2 million water wells in California that contribute to irregularity of groundwater and affecting the soil moisture. They range from hand-dug, shallow wells to carefully designed large-production wells drilled to great depths (California Dept. of Water Resources, 2018). More data is needed before we can recognize further attributions to low correlation between VIC and SMAP in that region. While systematic biases do not get revealed in correlations, the temporal consistency among the time series is captured.

1. Comments from referee
   Bottom of page 8. . .what exactly is meant by raw SMAP retrievals? Also, the list here seems to contain 2 (as stated in the text). Finally, the exact link between these 6 comparisons and plotted results in Figure 3 is a bit unclear. A couple more explanatory sentences would help here.

2. Author's response
   We fixed the number from 6 to 2 to avoid confusion and over explaining the details. Raw SMAP is SMAP Level 3, and the word raw is removed.

3. Author's changes in manuscript
   We used the KS test for each grid, comparing the modeled beta distribution of the long-term VIC with the modeled beta distribution of the short-term VIC, in both warm and cold seasons. This shows if the long-term and short-term distributions are statistically indistinguishable.

1. Comments from referee
   Bottom of page 13/of page 14. It is not clear to me how the SMAP L4 product could possibly detect the impact of groundwater extraction (using a land model which does not consider the impact of well pumping on saturated zone calculations and assimilation observations sensitive to only the top 5 cm of the soil column). Therefore, the attribution presented here seems potentially misguided. This discussion should be either strengthened or removed.

2. Author's response
   Since this is beyond the scope of the paper we decided to remove this argument from the paper.

3. Author's changes in manuscript
   NA

1. Comments from referee
   The abstract spends too much time discussing SMAP background (in the first paragraph) and too little time defining the contribution of this particular manuscript (see major point above).

2. Author's response
   We agree with that, now the abstract is revised to embody more informative aspects of the information provided in the paper.

3. Author's changes in manuscript
   Abstract: Since April 2015, NASA's Soil Moisture Active Passive (SMAP) mission has monitored near-surface soil moisture, mapping the globe (between $85.044°N/S$) using an L-band (1.4 GHz) microwave radiometer in 2-3 days depending on location. Of particular interest to SMAP-based agricultural applications is a monitoring product that assesses the SMAP near-surface soil moisture in terms of probability percentiles for dry and wet conditions. However, the short SMAP record length poses a statistical challenge for meaningful assessment of its indices. This study presents initial insights about using SMAP for monitoring drought and pluvial regions with a first application over the Contiguous United States (CONUS). SMAP soil moisture data from April 2015 to December 2017 at both near-surface (5cm) SPL3SMP, or Level 3, at ∼36 km resolution; and root zone SPL4SMAU, or Level 4, at ∼9 km resolution were fitted to beta distributions and were used to construct probability distributions for warm (May-October) and cold (November-April) seasons. To assess the data adequacy and have confidence in using short-term SMAP for drought index estimate, we analyzed individual grids by defining two filters and a combination of them, which could separate 5,815 grids covering CONUS into passed and failed grids. The two filters were: (1) The Kolmogorov-Smirnov (KS) test for beta-fitted long-term and short-term Variable Infiltration Capacity (VIC) LSM with 95% confidence; and (2) Good correlation ($\geq 0.4$) between beta-fitted VIC and beta-fitted SPL3SMP. To evaluate which filter is the best, we defined a Mean Distance ($MD$) metric, assuming VIC index at 36 km resolution is the ground truth. For both warm and cold seasons, the union of the filters – which also gives the best coverage of the grids throughout CONUS – was chosen to be the most reliable filter. We visually compared our SMAP-based drought index maps with metrics such as U.S. Drought Monitor (from D0-D4), SPI 1 month and VIC near surface from Princeton University. The root zone drought index maps were shown to be similar to those produced by the VIC at root zone, SPI 3 month, and GRACE. This study is a step forward towards building a national and international soil moisture monitoring system, without which, quantitative measures of drought and pluvial conditions will remain difficult to judge.

1. Comments from referee
   The SMAP product version names in the manuscript differ from the official product names/acronyms (see https://smap.jpl.nasa.gov/data/). . .good to use the official versions.

2. Author's response
   We used the product names from NSIDC (https://nsidc.org/data/smap/smap-data.html) and not the names from NASA. Since people should go to NSIDC, that naming seems to the best. The inconsistency between JPL and NSIDC is a problem.

3. Author's changes in manuscript
   NA

1. Comments from referee
   Page 3/Line 20. . .double parentheses.

2. Author's response
   That is fixed!

3. Author's changes in manuscript
   The approach (Sheffield et al., 2004) took was to fit the VIC-simulated soil moisture to probability distributions, usually beta distributions, where the percentiles are translated to the index values that range from 0 to 1.

1. Comments from referee
   Page 7/Line 4. . .better to say too tightly bounded.

2. Author's response
   This whole paragraph is reworded.

3. Author's changes in manuscript
   Please read the next comment's Author's changes.

1. Comments from referee
   Page 7/Lines 9-11. . .reword to clarify. . .unclear how the moment matching approach applied here differences from that of Sheffield et al. (2004).

2. Author's response
   We made the explanation clearer under Fitting the beta distribution to the SMAP time series section.

3. Author's changes in manuscript
   There are, however, differences in our approach from that in Sheffield et al. (2004). Firstly, the basis of the data used in Sheffield et al. (2004) was simulated soil moisture from VIC while ours is remotely sensed data. Secondly, to calculate the bounds of beta distribution [a, b], Sheffield et al. (2004) used the first (last) 10% of the sorted soil moisture values linearly related to the empirical cumulative distribution function. In our study, this approach did not yield useful results with the estimated limits for a (b) for SMAP, often did not cover the full range of observed values, preventing interpretation of the historical data. Our methodology for obtaining beta distribution parameters a and b is discussed in this section.

1. Comments from referee
   Figure 5 needs a color key. . .not clear what grey shading indicates.

2. Author's response
   We have explained this in the caption and furthermore, we have a Figure 6 now that includes the information of Figure 5 in it and it is color coded. To avoid confusion with different colors codes in Figure 5 and Figure 6 and redundancy of the same numbers, we explained what gray area is in the caption of Figure 5.

3. Author's changes in manuscript
   Please see Figure 6 and the caption of 5.

1. Comments from referee
   Bottom of page 12. . .where exactly is this grid analysis presented? Unclear what is being referred to here.

2. Author's response
   We removed grid analysis phrase to avoid confusion and rephrased it.

[revised manuscript text omitted]

---

## Author Response (AR1)

**Response to Referee #1**

Paper's New Title: Developing a drought monitoring index for the Contiguous U.S. using SMAP

**September 10, 2018**

**1. Comments from referee**

With interest I have read the manuscript entitled "A SMAP-Based Drought Monitoring Index for the United States", it is interesting and well written (although I am not a native English speaker!). It is certainly of interest for HESS readers. The manuscript details a new SMAP-based index for drought monitoring over the Continental US (although the title mentions [...] the United States).

**2. Author's response**

Thank you for this comment. We have changed the title to the one above. We believe is targeting the content of the paper.

3. Author's changes in manuscript

Paper's New Title: Developing a drought monitoring index for the Contiguous U.S. using SMAP

1. Comments from referee

The methodology stems on previous work from Sheffield et al., 2004 and is applied to the recent SMAP data. The resulting drought index is then compared to other already existing index like SPI-1&3, and another one GRACE-based. While the article is clear (at least to me), I am missing some more analyses of the drought Index for the manuscript to go from [...] a demonstration of the reliability [...] with lot of text to a proper journal article (There is no results section?).

2. Author's response

There are some differences between our approach and those from Sheffield et al., 2004. After carefully revisiting there paper, we made major numerical analysis on the added value of the drought index using SMAP and expanded our Results section. To assess the data adequacy and have confidence in using short-term SMAP for drought index estimate, we analyzed individual grids by defining two filters and a combination of them, which could separate 5,815 grids covering CONUS into passed and failed grids. The two filters were: (1) The Kolmogorov-Smirnov (KS) test for beta-fitted long-term and short-term Variable Infiltration Capacity (VIC) LSM with 95% confidence; and (2) Good correlation (0.4) between beta-fitted VIC and beta-fitted SPL3SMP. To evaluate which filter is the best, we defined a Mean Distance (MD) metric, assuming VIC index at 36 km resolution is the ground truth. The new reliability analysis is described under Data adequacy filter section.

3. Author's changes in manuscript

The approach selected here is somewhat similar to that from Sheffield et al. (2004) where the soil moisture time series are fit to a beta distribution (with upper and lower bounds) and the distribution percentiles are the index values. There are, however, differences in our approach from that in Sheffield et al. (2004). Firstly, the basis of the data used in Sheffield et al. (2004) was simulated soil moisture from VIC while ours is remotely sensed data. Secondly, to calculate the bounds of beta distribution [a, b], Sheffield et al. (2004) used the first (last) 10% of the sorted soil moisture values linearly related to the empirical cumulative distribution function. In our study, this approach did not yield useful results with the estimated limits for a (b) for SMAP, often did not cover the full range of observed values, preventing interpretation of the historical data. Our methodology for obtaining beta distribution parameters a and b is discussed in this section.

Figure 1: (a) Correlations (R) between VIC and SMAP beta models for the warm season (average R=0.57) and (b) cold season (average R=0.56). White regions signify negative correlation.

**0.1 Data Adequacy Filters**

Insufficient SMAP record length may result in unreliable index values. To be meaningful in using short SPL3SMP data for making confident predictions, we will analyze which grids have the highest certainty in our SMAP drought index. That is, we perform adequacy analysis, and defining filters that separate grids with high reliability in drought monitoring and prediction from ones where we dont expect our predictions to be as accurate. We first define two filters which can separate the 5,815 grids covering CONUS into grids that passed and failed quality control. The two filters are:

- 1. The Kolmogorov-Smirnov (KS) test for beta-fitted long-term and short-term VIC with 95% confidence;
- 2. Good correlation ( $\geq 0.4$ ) between beta-fitted VIC and beta-fitted SPL3SMP.

Below we expand upon these two filters and then show how we used them to numerically find the best SPL3SMP filter. We also investigate if combinations of the filters are superior to the individual filters taken alone.

**0.1.1 Kolmogorov-Smirnov (KS) filter**

The KS test is a well-known nonparametric statistical test that compares whether two samples are coming from the same continuous distribution. We used the KS test for each grid, comparing the modeled beta distribution of the long-term VIC with the modeled beta distribution of the short-term VIC, in both warm and cold seasons. This shows if the long-term and short-term distributions are statistically indistinguishable. If this strong condition is satisfied for a grid, then it is reasonable to assume for that grid that the short SMAP time series would be consistent with a hypothetical long SMAP time series. The null hypothesis – that the underlying beta distribution of short-term soil moisture data is the same as the underlying beta distribution of long-term soil moisture data for VIC – is rejected for values of the KS statistic D that exceed a critical value at the 95% significance level:  $D_{critical} = \frac{1.36}{\sqrt{n}}$  where n is the number of observed variable (Lindgren, 1962). Figure 2 shows which grids passed the 95% KS test: there, we have confidence that the SMAP drought (pluvial) indices provide reliable risk levels given the current period of record.

**0.1.2 Correlation Filter**

As mentioned earlier, one of the key assumptions of this paper is that if the beta distribution fit to the shortterm VIC series is statistically consistent with beta fit to the long-term VIC time series, then we assume that the short-term beta-fitted SMAP series is consistent with the hypothetical long-term beta-fitted SMAP time series. This is possible because VIC modeled soil moisture is validated by ground measurements (Pan et al.)

---

## Author Response (AR2)

Title: Developing a drought monitoring index for the Contiguous U.S. using SMAP
S. Sadri, Ming Pan, E.F. Wood

Author's response:

Section 2.1, p. 5: Clarify what product version of the SMAP SPL3SMP data is used here. In the latest SMAP product release (R16) frozen ground or snow-covered areas are masked out using a combined NPR and single-channel algorithm based freeze-thaw retrieval, rather than just NPR (Line 10). The NPR is predominantly used at high latitudes where there's a larger V-H Tb difference and sufficient NPR signal-noise, whereas the single channel algorithm is predominantly used at lower temperate latitudes and over vegetated areas.

The following explanation is added:
For this study, version 4 of SPL3SMP is used, which is the release version from the very beginning of the launch of SMAP. The release number changes over time. R16 version is the latest version released in June 2018. However, in all release versions of SMAP, including version 4, regions with permanent snow/ice, frozen ground, excessive static or transient open water in the cell or excessive radio-frequency interference (RFI) in the sensor data, and heavy vegetation (vegetation water content > 4.5 kg/m^2) are masked out using a Normalized Polarization Ratio (NPR)-based passive freeze-thaw retrieval.

p. 5, Ln 23 (grammar): "…without the discontinuous data.." should be "…without discontinuous data restrictions due to gaps in the SPL3SMP soil moisture retrievals".

This is fixed now.

p. 7, Ln 7 (grammar): "..of beta.." should be ".. of the beta..".

This is fixed now.

p. 8, Figure 2 caption (grammar): "..bottom row: as the .." should be "..bottom row: same as the…".

This is fixed now.

p. 11, Ln 24: Include Lawston et al. 2017. GRL reference here to justify suspect irrigated lands impact on SMAP soil moisture retrievals.
https://agupubs.onlinelibrary.wiley.com/doi/full/10.1002/2017GL075733

Reference is included now.

Results discussion of Figure 5: Can the authors provide a potential reason for the low warm season correspondence in the PNW region? I would also expect to see more of a pattern of low

correspondence over the major mountain areas (e.g. Rockies, Sierra-Cascades) given the coarse SMAP Tb footprint, but this isn't very apparent; can the authors talk to this as well?

The following explanation has been added:
In the warm season, the majority of the grids whose underlying short term and long term beta distribution were different were in the western U.S. The low warm season correspondence in the Pacific Northwest region is particularly apparent. The PNW region is covered by dense forests, mountains, and heavily regulated agricultural lands by irrigation. This contributes to the fact that most grids in PNW do not pass the KS filter.  A pattern of low correspondence over the major mountain areas (e.g. Rockies, Sierra-Cascades) is also apparent, given the coarse SMAP brightness temperature (Tb) footprint and dense vegetation.

Results discussion of Figure 6: Can the authors include an additional explanation of the low warm season correspondence in the great lakes region? I suspect this may be due to contamination of SMAP Tb and soil moisture retrievals from small water bodies, since the soil moisture algorithm.

I think you are mistaking the low correlation between VIC and SMAP during the cold season in the great lakes with KS test. As Figure 6 shows in cold season, the KS test has better correspondence than the correlation. Nevertheless, I added the following explanation:
The Great Lakes region, Minnesota, and Mid Atlantic Region do not show a high correlation between VIC and SMAP in the cold season. Snow, heavy canopy, and land development cause SMAP retrievals to have errors. In addition, this region does not have a good coverage of soil moisture and has less number of retrievals per grid (Figure \ref{fig:overpass}).

Figure 6-7 captions: Clarify what the black x symbols are in the SPL3SMP maps.

Thank you! The explanation has been added as: The black x symbols in the SPL3SMP maps are the grids that passed neither filter and were shown as white grids in Figure 6.

p. 16 bottom (sentence structure): "..SMAP to offer index.." should be "..SMAP index to offer.." .

It's fixed now.

p. 17 bottom (sentence structure): "Our index SPL3SMP index product maps…" should be "Our SPL3SMP index maps…".

It's fixed now.

p. 17 bottom: More information and clarification is needed regarding the use of the GRACE NDMC product for the 2017 period. The NDMC product is a model data assimilation product that combines GRACE data with other meteorological information. In 2017 the GRACE sensor was failing and the resulting water storage observations were unreliable. As such, the last GRACE

gravity field retrievals and RL06 reprocessing only go through June 2017 and Aug 2016, respectively. Figure 9 and the associated results and discussion therefore likely don't reflect actual GRACE observations for 2017.

Thank you for mentioning that. In fact, the data from GRACE does look different from the other products, especially on 2017-10-17. The following explanation has been incorporated into the text:
Both SPL3SMP and SPL4SMAU index maps seem to catch this drought event, although the event was more pronounced in the root zone than the surface. The maps of these two figures are also in general agreement. It is important to clarify that for 2017 period, the GRACE sensor was failing and the resulting water storage observations were unreliable. Therefore, the last GRACE gravity field retrievals processing only go through June 2017. Therefore, GRACE NDMC results associated with Figure \ref{tbl:rz_2017} are not consistent with other products and likely do not reflect actual GRACE observations for 2017.